# Steerable Transformers for Volumetric Data

**Soumyabrata Kundu** [1]   **Risi Kondor** [1,2]

## Abstract

We introduce Steerable Transformers, an extension of the Vision Transformer that is equivariant to the action of the Special Euclidean group $\mathrm{SE}(d)$. We propose an steerable self-attention mechanism that operates on features extracted by steerable convolutions. Our experiments in both two and three dimensions show augmenting steerable convolutional networks with steerable transformer leads to improved performance.

## 1. Introduction

Transformers have emerged as the preferred architecture for natural language processing tasks, with recent powerful models like Chat-GPT employing this framework. Their relatively straightforward design, coupled with remarkable success, has led to their widespread adoption across various domains, including image classification (Dosovitskiy et al., 2021), object detection (Carion et al., 2020), and graph based problems (Dwivedi & Bresson, 2020). The self-attention mechanism (Bahdanau et al., 2014) employed in transformer architectures has proven to be crucial for capturing relationships between different parts of the input sequence. Dosovitskiy et al. (2021) introduced transformers as an alternative to traditional convolutional architectures for vision tasks. Unlike convolutional neural networks (CNNs), which focus on local neighborhoods, transformers excel at capturing relations across different parts of the input.

Equivariant neural network architectures have gained significant popularity in recent years due to their inherent ability to comprehend the underlying symmetries of problems, making them highly effective tools for real world tasks. For instance, achieving equivariance to the permutation group $S_n$ is crucial in graph based problems, since the structure of the graph remains invariant under permutations of node labels (Thiede et al., 2020; Wang et al., 2020). Similarly, in vision tasks, it is desirable to have equivariance to rigid body transformations such as rotations and translations. Recent studies by Cohen & Welling (2016; 2017); Worrall et al. (2017); Weiler et al. (2018a); Weiler & Cesa (2019); Thiede et al. (2020); Anderson et al. (2019), and many more, have demonstrated the remarkable efficacy of these equivariant architectures in enforcing symmetry, without relying on brute force techniques like data augmentation.

In vision tasks, steerable convolutions have proven to be a powerful tool for enforcing equivariance to rotations and translations (Cohen & Welling, 2016; Worrall et al., 2017; Cohen & Welling, 2017; Cohen et al., 2018; Weiler et al., 2018a; Weiler & Cesa, 2019). These networks primarily operate in Fourier space, leveraging representations of the underlying group to extract rotation and translation equivariant features from the data. These architectures are effective at learning relevant features in local neighborhoods. Conversely, transformer architectures excel at learning relationships between distant regions of an image. Our contribution lies in combining these two concepts, yielding equivariant architectures that are capable of capturing both local and global patterns in images.

### 1.1. Related Work

The concept of attention was first proposed by Bahdanau et al. (2014) in the context of sequence modeling. Building on this, Vaswani et al. (2017) developed the transformer architecture for natural language translation tasks. Later, Dosovitskiy et al. (2021) extended the transformer framework to visual domains, proposing a hybrid model that integrates features extracted from conventional CNNs. In contrast, Ramachandran et al. (2019) introduced a fully attention based architecture that entirely replaces traditional convolutional networks.

Transformer architectures are inherently nonlinear in their inputs, and applying them directly into equivariant architectures can break equivariance. This has motivated increasing interest in developing transformer models that are designed to preserve equivariance. Romero & Cordonnier (2021) demonstrated that vision transformers using relative positional encoding (Shaw et al., 2018) exhibit translation

---

[1]Department of Statistics, University of Chicago, Chicago, USA [2]Department of Computer Science, University of Chicago, Chicago, USA. Correspondence to: Soumyabrata Kundu <soumyabratakundu@uchicago.edu>.

*Proceedings of the 42nd International Conference on Machine Learning*, Vancouver, Canada. PMLR 267, 2025. Copyright 2025 by the author(s).

equivariance. They extended the attention based framework of Ramachandran et al. (2019) by formulating the attention mechanism over the group $SE(2)$, thereby achieving both rotation and translation equivariance in two dimensions. Building on this approach, Xu et al. (2023) proposed a novel positional encoding operator to further improve performance. Additionally, Romero et al. (2020) incorporated attention directly into convolution by defining the attention scores over the equivariance group.

In three dimensions, attention based methods that enforce equivariance to rotations and translations have been explored. However, to the best of our knowledge, these efforts have been limited to point cloud data. Fuchs et al. (2020) introduced an $SE(3)$-equivariant architecture for point clouds by incorporating the self-attention mechanism into the Tensor Field Network (Thomas et al., 2018). Liao & Smidt (2023) proposed Equiformer, an equivariant extension of the Graph Attention Network (Veličković et al., 2018) for 3D point clouds, and subsequently, Liao et al. (2024) presented EquiformerV2, a more computationally efficient variant that reduces $SO(3)$ convolutions to $SO(2)$, leveraging ideas from Passaro & Zitnick (2023). Chen & Villar (2022) achieved equivariant attention by projecting data onto $SE(3)$-equivariant and invariant features, based on equivariant feature maps developed by Villar et al. (2021). Hutchinson et al. (2021) proposed a transformer framework with attention mechanism that is equivariant under the action of general Lie groups, using regular representations. For shape reconstruction tasks, Chatzipantazis et al. (2023) introduced an $SE(3)$-equivariant attention model.

### 1.2. Our Contribution

Romero et al. (2020) explored equivariant transformer architectures for two dimensional images by integrating the self-attention mechanism within convolutional layers. While effective, this comes at the cost of significantly higher memory usage. In contrast, the architectures introduced by Xu et al. (2023), which build on the model of Ramachandran et al. (2019), entirely eliminate convolutional components in favor of self-attention based mechanisms. Nevertheless, as noted by Xiao et al. (2021), incorporating a convolutional encoder prior to the transformer leads to better performance, suggesting a more balanced and effective design.

In three dimensions, Fuchs et al. (2020); Hutchinson et al. (2021); Liao & Smidt (2023) proposed $SE(3)$-equivariant transformer architectures tailored for point cloud data. Although volumetric data, in principle, can be conceptualized as point clouds arranged on a regular grid, using these point cloud based methods on dense volumetric inputs can impose inappropriate inductive biases, as it overlooks the structured nature of the underlying grid. Furthermore, treating volumetric data as point clouds and applying these methods can lead to excessive memory usage, often resulting in out-of-memory errors.

To address these challenges, we present a novel steerable transformer architecture specifically designed for *volumetric data* in $d$ dimensions. Our proposed steerable transformer integrates on top of a steerable convolutional encoder and, as demonstrated through our experiments, enhances the performance of steerable convolutions. Drawing inspiration primarily from Dosovitskiy et al. (2021) and Fuchs et al. (2020), our method operates in the Fourier domain, by using group representations to learn equivariant features. While Vaswani et al. (2017) employ fixed positional functions and Dosovitskiy et al. (2021) use learnable positional encodings, our method integrates both ideas by modulating a fixed positional function with learnable scaling parameters.

## 2. Background

In this section we introduce the relevant background required to design the steerable transformer architecture.

### 2.1. The Multihead Self-Attention Mechanism

The attention mechanism, originally introduced by Bahdanau et al. (2014) for sequence-to-sequence learning, relies on three core components for each element in the input sequence: query vector $q_i$, key vector $k_j$, and value vector $v_j$. The scores $s_{ij}$ measure the compatibility between the query and key vectors using a scaled dot product. These scores are passed through a softmax function to produce attention weights $\alpha_{ij}$, which quantify the contribution of each value vector to the output. The output is then computed as a weighted sum of the value vectors:

$$\text{ATTN}(q_i, \{k_j\}, \{v_j\}) = \sum_{j=1}^{N} \alpha_{ij} v_j.$$

The attention weights $\alpha_{ij}$ are given by

$$s_{ij} = \frac{q_i^T k_j}{\sqrt{d_K}}, \quad \alpha_{ij} = \frac{\exp(s_{ij})}{\sum_{j'=1}^{N} \exp(s_{ij'})},$$

where $d_K$ denotes the dimension of the query and key vectors, and $N$ represents the number of elements in the input sequence. Vaswani et al. (2017) proposed a Multihead version of this attention mechanism which involves stacking multiple attention mechanisms, concatenating their outputs and applying a linear transformation to get the final result:

$$\text{Concat}(\text{ATTN}^1, \dots, \text{ATTN}^h) W_O, \quad W_O \in \mathbb{R}^{h d_V \times d_{\text{model}}}.$$

Here, $d_V$ denotes the dimension of the value vectors, and $d_{\text{model}}$ denotes the dimension of the output. Typically, the

query, key, and value vectors are obtained by multiplying the input sequence with learnable weight matrices followed by adding *positional encoding*.

**Positional Encoding:** Positional encoding in transformers is used to embed information about the position of words or tokens within a sequence into their input representations. In vision applications, these encodings help the model retain spatial awareness by indicating the original locations of image patches. Vaswani et al. (2017) used sine and cosine functions at different frequencies to encode positions. A refinement of this approach, known as relative positional encoding, was proposed by Shaw et al. (2018). Unlike fixed encodings, this method learns representations that capture relative distances between elements in the input. As we will discuss later, incorporating relative positional encoding is essential for preserving equivariance.

## 2.2. Vision Transformers

Dosovitskiy et al. (2021) adapted the transformer architecture for vision tasks. Their method involves partitioning an image into equal sized patches, which are then processed by a transformer. The resulting feature representations are used for downstream tasks like image classification. Assuming a single input channel, the operations of a vision transformer layer can be summarized as follows:

$$z_0 = [\mathbf{x}_1 E, \mathbf{x}_2 E, \ldots, \mathbf{x}_N E] + E_{\text{pos}}$$
$$z'_m = \text{MULTI-ATTN}(\text{LN}(z_{M-1})) + z_{M-1} \quad m = 1, \ldots M$$
$$z_m = \text{MLP}(\text{LN}(z_{m-1})) + z'_m \quad m = 1, \ldots M.$$

Here, $\mathbf{x}_i \in \mathbb{R}^p$ represents a flattened image patch, $E \in \mathbb{R}^{p \times d_{\text{model}}}$ is the learnable linear weights matrix, $E_{\text{pos}} \in \mathbb{R}^{N \times d_{\text{model}}}$ is the positional encoding, and $m$ indexes the various blocks of the transformer. LN denotes a normalization layer, and MLP stands for a Multilayer Perceptron. $d_{\text{model}}$ is the dimension of the features fed into the transformer, which remains fixed throughout the entire transformer layer. Typically, the MLP layer comprises two hidden layers separated by a non-linearity (Vaswani et al., 2017; Dosovitskiy et al., 2021).

## 2.3. Equivariance

Equivariance describes a property in which a map between vector spaces commutes with the action of a group $G$ (see Appendix A.2 for definition of group action). Specifically, for each element $g \in G$, suppose we have linear operators $T_g : \mathcal{V} \to \mathcal{V}$ and $T'_g : \mathcal{W} \to \mathcal{W}$ that represent the action of $g$ on vector spaces $\mathcal{V}$ and $\mathcal{W}$, respectively. A map $\lambda : \mathcal{V} \to \mathcal{W}$ is said to be equivariant if, for all $\mathbf{v} \in \mathcal{V}$ and $g \in G$,

$$\lambda(T_g(\mathbf{v})) = T'_g(\lambda(\mathbf{v})).$$

Intuitively, this means that applying the group action before or after the map $\lambda$ leads to the same outcome, guaranteeing that the resulting features respect the symmetry structure imposed by the group. Equivariance naturally arises as a desirable property in many real world applications that involve underlying symmetries, such as permutations in graph structured data or rigid body transformations in images. There is extensive research on equivariant neural networks, particularly in the setting where the map $\lambda$ is linear (Worrall et al., 2017; Cohen et al., 2018; Weiler et al., 2018a; Cohen & Welling, 2017; 2016; Weiler & Cesa, 2019; Kondor et al., 2018; Anderson et al., 2019; Thomas et al., 2018).

## 2.4. Steerable Convolutions

Steerable neural networks are designed to exhibit equivariance to the action of the Special Euclidean group $\text{SE}(d)$ in $d$ dimensions (see Appendix A.1 for definition). Steerable convolutions represent the most general class of linear maps that are equivariant with respect to the action of $\text{SE}(d)$ (Kondor & Trivedi, 2018). Although practical neural networks work with finite resolution rasterized images, we adopt a continuous formulation for the network's inputs and filters to simplify the mathematical exposition and provide a clear overview of steerable convolutions. In this setting, a $d$ dimensional image with a single input channel is represented as a compactly supported function $f^{\text{in}} : \mathbb{R}^d \to \mathbb{R}$, and analogously, a filter is also represented as a compactly supported function $w : \mathbb{R}^d \to \mathbb{R}$. The output of a steerable convolution layer is a function $f^{\text{out}} : \text{SE}(d) \to \mathbb{R}$ that can be expressed as

$$f^{\text{out}}(\mathbf{x}, R) = \int_{\mathbb{R}^d} f^{\text{in}}(\mathbf{x} + R\mathbf{y}) \, w(\mathbf{y}) \, d\mathbf{y}, \qquad (1)$$

$$f^{\text{out}}(\mathbf{x}, R) = \int_{\text{SO}(d)} \int_{\mathbb{R}^d} f^{\text{in}}(\mathbf{x} + R\mathbf{y}, RR')$$
$$w(\mathbf{y}, R') \, d\mathbf{y} \, d\mu(R'). \qquad (2)$$

Here, $\mathbf{x} \in \mathbb{R}^d$, $R \in \text{SO}(d)$ and $\mu$ denotes the Haar measure on $\text{SO}(d)$ (see Appendix A.4 for definition). The presence of two distinct formulas is due to the fact that the input to the first layer is a vector field on $\mathbb{R}^d$, whereas the input to subsequent layers is a vector field on $\text{SE}(d)$. Equation (1) represents the formula for the first layer, while (2) represents subsequent layers. These layers are equivariant under the action of the group $\text{SE}(d)$. Specifically, if the input function $f^{\text{in}}$ is transformed by an element $(\mathbf{t}, R) \in \text{SE}(d)$ according to

$$[(\mathbf{t}, R)^{-1} \cdot f^{\text{in}}](\mathbf{x}) = f^{\text{in}}(R\mathbf{x} + \mathbf{t}), \qquad (3)$$

then the corresponding output $f^{\text{out}}$ transforms as

$$[(\mathbf{t}, R)^{-1} \cdot f^{\text{out}}](\mathbf{x}, R') = f^{\text{out}}(R\mathbf{x} + \mathbf{t}, RR'). \qquad (4)$$

For compact groups like $SO(d)$, where discretization is not straightforward, it is advantageous to compute the rotation component of the output feature map of (1)-(2) in Fourier space (Worrall et al., 2017). The Fourier transform of a function $f$ on a compact group can be computed by integrating it against it's irreducible representations or *irreps* (see Appendix A.3 for definition):

$$\widehat{f}(\rho) = \int_{SO(d)} f(R)\, \rho(R)\, d\mu(R). \tag{5}$$

Leveraging the Convolution Theorem, equations (1)-(2) in Fourier space becomes a matrix product of the Fourier transforms of the input and the filter. By the translation property of the Fourier transform, the vector fields satisfy the property that when the input transforms under the action of $SE(d)$ as in (3), the corresponding output transforms as

$$[(\mathbf{t}, R)^{-1} \cdot \widehat{f^{\text{out}}}](\mathbf{x}, \rho) = \rho(R)^{\dagger} \widehat{f^{\text{out}}}(R\mathbf{x} + t, \rho). \tag{6}$$

In the two dimensions, the irreps of $SO(2)$ are just complex exponential indexed by integers, $\rho_k(\theta) = e^{\iota k\theta}$ for $k \in \mathbb{Z}$. Therefore, the $k^{\text{th}}$ frequency component of the output feature map is a function

$$\widehat{f^{\text{out}}}(\cdot, k) : \mathbb{R}^2 \to \mathbb{C}. \tag{7}$$

In three dimensions, the irreps of $SO(3)$ are indexed by non-negative integers $\ell \in \mathbb{Z}_{\geq 0}$, and are given by $\rho_\ell(R) = D^\ell(R)$, the so called Wigner D-matrices (Wigner, 1932), which are unitary matrices of size $2\ell+1$. Since the columns of the output can be interpreted as separate channels, the $\ell^{\text{th}}$ Fourier component of the output feature map for a single channel can be represented as a function

$$\widehat{f^{\text{out}}}(\cdot, \ell) : \mathbb{R}^3 \to \mathbb{C}^{2\ell+1}. \tag{8}$$

**Non-linearities:** In steerable neural networks, applying non-linearities like ReLU in Fourier space can disrupt equivariance. One approach to mitigate this issue is to transition back to time domain, apply the desired non-linearity there, and then return to Fourier domain (Cohen et al., 2018). However, this back and forth transformation can be computationally expensive and error prone, especially on groups like $SO(3)$, where the uniform grid gives rise to singularities.

As an alternative, Fourier space non-linearities are preferred in steerable neural networks. These non-linearities act directly on Fourier space while preserving equivariance. Several such non-linearities can be found in the literature. Worrall et al. (2017) apply non-linearity to the norm of the Fourier vector, as the norm remains invariant under rotations. Kondor et al. (2018); Anderson et al. (2019) use the Clebsch–Gordan non-linearity, which involves a tensor product of Fourier vectors followed by a Clebsch–Gordan decomposition. Weiler et al. (2018a) introduce another steerable convolution filter to serve as a non-linearity. These methods enable effective non-linear transformations in Fourier space, while preserving equivariance.

## 3. Method

The methodology behind steerable transformers builds on the foundation of vision transformers (Dosovitskiy et al., 2021), which operate on "patchified" images, i.e., linear projections of fixed sized patches extracted from the input image. Conceptually, this patchification process is analogous to applying strided convolutions, where the stride is equal to the kernel size. Dosovitskiy et al. (2021) also propose a hybrid architecture that extends this idea by employing a deeper convolutional encoder to extract more expressive features, which are then used as input to the transformer layers. In steerable transformers, we build on this idea by using feature maps extracted from steerable convolution encoder as input to the transformer architecture.

In the discussion of steerable convolutions in Section 2.4, the input, filter and consequently, the output were modeled as compactly supported functions defined over continuous domains. We adopt this continuous perspective in the discussion that follows, while acknowledging that, in practical implementations, the output of a steerable convolutional encoder for a given irrep is represented as a collection of complex-valued vectors arranged on a discrete grid. Accordingly, any integrals in the theoretical formulation can be approximated by summations over this grid. While the number of channels associated with each irrep may vary in practice, for brevity, we consider the simplified case in which all irreps share the same number of channels. The generalization to a variable channel setting is straightforward.

Assume that the input has $d_{\text{model}}$ channels for each irrep. Then, the input to the transformer corresponding to an irrep $\rho$ can be represented as a function

$$f^{\text{in}}(\mathbf{x}, \rho) \in \mathbb{C}^{d_\rho \times d_{\text{model}}}, \tag{9}$$

which has compact support $\Omega(f^{\text{in}})$ for all $\rho$, and satisfies the equivariance constraint (6). Here, $\mathbf{x} \in \mathbb{R}^d$, and $d_\rho$ denotes the dimension of $\rho$. In two dimensions, all irreps are one dimensional (7), whereas in three dimensions, the $\ell^{\text{th}}$ representation has dimension $2\ell + 1$ (8).

### 3.1. Steerable Self-Attention Mechanism

The query, key, and value vectors are computed by applying learnable weight matrices to the input $f^{\text{in}}$ in (9), followed by the addition of positional encodings. Romero & Cordonnier (2021) showed that incorporating relative po-

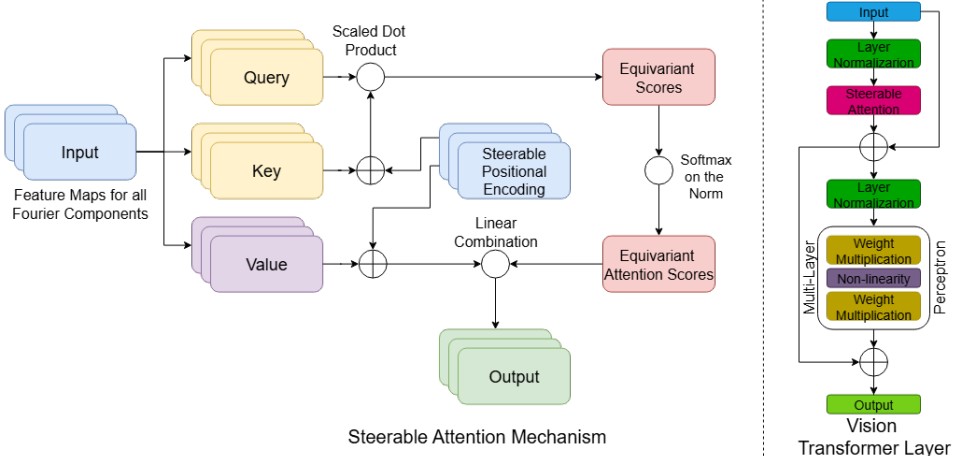

Figure 1: The schematic illustrates the steerable self-attention mechanism for a single head ($h = 1$) and one query dimension ($d_K = 1$) (left) and a steerable transformer encoder layer (right); c.f. Figure 1 by Dosovitskiy et al. (2021).

sitional encodings (Shaw et al., 2018) into self-attention enforces translation equivariance. Our design of *steerable positional encodings* is inspired by this relative positional encoding. We employ a hybrid strategy that blends learnable weights and fixed functions to encode positional information.

In the context of steerable neural networks, it is essential that positional encodings also encode rotational structure, which is naturally expressed in Fourier space. Accordingly, for each irrep $\rho$, we define the steerable positional encoding as a continuous function

$$\mathbf{P}^{(\rho)} : \mathbb{R}^d \times \mathbb{R}^d \to \mathbb{C}^{d_\rho}.$$

We elaborate on the exact forms of these encodings in the next section. For now, we focus on the design of the steerable self-attention mechanism. Within this framework, the query, key, and value vectors corresponding to an input map $f^{\text{in}}$ are given by

$$\mathbf{q}^{(\rho)}(\mathbf{x}) = f^{\text{in}}(\mathbf{x}, \rho)\mathbf{W}_Q^{(\rho)} \tag{10}$$

$$\mathbf{k}^{(\rho)}(\mathbf{x}, \mathbf{y}) = f^{\text{in}}(\mathbf{x}, \rho)\mathbf{W}_K^{(\rho)} + \mathbf{P}^{(\rho)}(\mathbf{x}, \mathbf{y}) \tag{11}$$

$$\mathbf{v}^{(\rho)}(\mathbf{x}, \mathbf{y}) = f^{\text{in}}(\mathbf{x}, \rho)\mathbf{W}_V^{(\rho)} + \mathbf{P}^{(\rho)}(\mathbf{x}, \mathbf{y}), \tag{12}$$

where $\mathbf{W}_Q^{(\rho)}, \mathbf{W}_K^{(\rho)} \in \mathbb{C}^{d_{\text{model}} \times d_K}$ and $\mathbf{W}_V^{(\rho)} \in \mathbb{C}^{d_{\text{model}} \times d_V}$. Next, the score function is calculated by taking the scaled dot product between the query and key vectors:

$$s(\mathbf{x}, \mathbf{y}) = \sum_\rho \frac{\text{vec}(\mathbf{q}^{(\rho)}(\mathbf{y}))^\dagger \text{vec}(\mathbf{k}^{(\rho)}(\mathbf{x}, \mathbf{y}))}{\sqrt{d_K}}. \tag{13}$$

Here, the vec operation flattens the matrix into a vector. The attention scores are then obtained by applying the softmax function to the raw scores. In continous formulation

this can be expressed as

$$\alpha(\mathbf{x}, \mathbf{y}) = \frac{\exp(|s(\mathbf{x}, \mathbf{y})|)}{\int_{\Omega(f^{\text{in}})} \exp(|s(\mathbf{x}, \mathbf{z})|)\, d\mathbf{z}}. \tag{14}$$

Since we are dealing with complex numbers, we use the absolute value of the scores, and the dot product involves the conjugate transpose, denoted by $\cdot^\dagger$, instead of a regular transpose. Finally, the output for each position and Fourier component is computed as a linear combination of the value vectors and attention scores:

$$f^{\text{out}}(\mathbf{x}, \rho) = \int_{\Omega(f^{\text{in}})} \alpha(\mathbf{x}, \mathbf{y})\mathbf{v}^{(\rho)}(\mathbf{x}, \mathbf{y})\, d\mathbf{y}. \tag{15}$$

It is easy to verify that if the steerable positional encodings are set to zero, then by the unitary nature of the irreps, the output of the self-attention mechanism satisfies the equivariance constraint (6). However, to encode the spatial information of the input patches, we want to use non-trivial functions as steerable positional encodings. The following theorem establishes the necessary and sufficient condition that $\mathbf{P}^{(\rho)}$ must satisfy for the self-attention mechanism to preserve equivariance.

**Theorem 1.** *For any $f^{\text{in}}$ that transforms under the action of $\mathrm{SE}(d)$ according to the equivariance constraint (6), the output of the self-attention mechanism also satisfies (6) for any collection of weight matrices if and only if, for every $(\mathbf{t}, R) \in \mathrm{SE}(d)$, the positional encodings $\mathbf{P}^{(\rho)}$ satisfy*

$$\mathbf{P}^{(\rho)}(R\boldsymbol{x} + \mathbf{t}, R\boldsymbol{y} + \mathbf{t}) = \rho(R)\,\mathbf{P}^{(\rho)}(\boldsymbol{x}, \boldsymbol{y}), \tag{16}$$

*for all $\boldsymbol{x}, \boldsymbol{y} \in \mathbb{R}^d$ and for all irreps $\rho$.*

In the following section, we elaborate on the construction of $\mathbf{P}^{(\rho)}$ that satisfies the constraint (16).

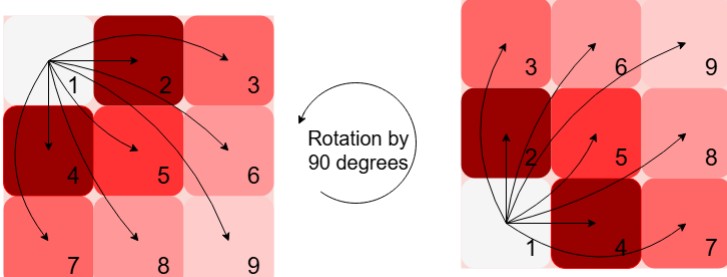

Figure 2: Visual representation of steerable positional encoding. Arrows denote directional components, while the color gradient indicates magnitude, decaying proportionally to $r^{-2}$.

In practice, the integrals appearing in equations (14) and (15) are approximated by discrete summations over a finite set of grid points. Specifically, the attention weights and output are computed as

$$\alpha(\mathbf{x}_i, \mathbf{x}_j) = \frac{\exp(|s(\mathbf{x}_i, \mathbf{x}_j)|)}{\sum_{j'=1}^{N} \exp(|s(\mathbf{x}_i, \mathbf{x}_{j'})|)}$$

$$f^{\text{out}}(\mathbf{x}_i, \rho) = \sum_{j=1}^{N} \alpha(\mathbf{x}_i, \mathbf{x}_j) \mathbf{v}^{(\rho)}(\mathbf{x}_i, \mathbf{x}_j),$$

respectively, where $\{\mathbf{x}_i\}_{i=1}^{N}$ denotes a set of discrete spatial locations on a grid. For multihead self-attention, this mechanism is repeated independently for $h$ heads. Finally, the outputs are concatenated, and each Fourier component is scaled by another matrix $\mathbf{W}_O^{(\rho)} \in \mathbb{C}^{hd_V \times d_{\text{model}}}$. Consistent with Vaswani et al. (2017), we choose $d_V = d_K = d_{\text{model}}/h$.

## 3.2. Steerable Positional Encoding

Theorem 1 establishes necessary and sufficient conditions that $\mathbf{P}^{(\rho)}$ must satisfy to ensure that the self-attention mechanism preserves equivariance. In particular, for any translation vector $\mathbf{t} \in \mathbb{R}^d$, the positional encoding must satisfy the condition

$$\mathbf{P}^{(\rho)}(\mathbf{x} + \mathbf{t}, \mathbf{y} + \mathbf{t}) = \mathbf{P}^{(\rho)}(\mathbf{x}, \mathbf{y}).$$

This implies that $\mathbf{P}^{(\rho)}$ depends only on the relative position, and can thus be expressed as a function of a single argument $\mathbf{P}^{(\rho)}(\mathbf{x} - \mathbf{y})$. As a result, the encoding is *invariant* under translations. Furthermore, for any rotation $R \in \text{SO}(d)$, it has the property that

$$\mathbf{P}^{(\rho)}(R\mathbf{x}) = \rho(R)\,\mathbf{P}^{(\rho)}(\mathbf{x}). \qquad (17)$$

A natural and expressive choice for such a function is the Spherical Harmonic basis functions (Frye & Efthimiou, 2012). In two dimensions, this is takes the form

$$P^{(k)}(\mathbf{x}) = \phi(r, k)e^{ik\theta}, \quad \mathbf{x} = \begin{bmatrix} r\cos\theta \\ r\sin\theta \end{bmatrix}, \qquad (18)$$

for $k \in \mathbb{Z}$. Here, $\phi(r, k)$ modulates the encoding strength for nearby and distant points. Similarly, in three dimensions, we have

$$P^{(\ell)}(\mathbf{x}) = \phi(r, l)Y^{(\ell)}(\theta, \phi), \quad \mathbf{x} = \begin{bmatrix} r\sin\theta\cos\phi \\ r\sin\theta\sin\phi \\ r\cos\theta \end{bmatrix}, \qquad (19)$$

for $\ell \in \mathbb{Z}_{\geq 0}$. Here $Y^{(\ell)}$ denotes the spherical harmonics of the unit sphere $\mathbb{S}^2$ in three dimensions (Byerly, 1893). The steerability property of positional encoding (17) implies $\mathbf{P}^{(\rho)}(0) = 0$ for any $\rho$ except for the constant representation ($k = 0$ and $\ell = 0$ in two and three dimensions, respectively), since $\mathbf{P}^{(\rho)}(0) = \rho(R)\mathbf{P}^{(\rho)}(0)$ for all $R \in \text{SO}(d)$. This condition can be enforced by setting $\phi$ to zero when $r = 0$. For our experiments, we used

$$\phi(r, \rho) = w_\rho r^{-2} \mathbb{1}_{r>0},$$

where $w_\rho$ is a learnable scalar. This kind of inverse square modulation of the radial component results in positional encoding assigning higher weight to neighboring points, and the weights decrease as points move further apart (see Figure 2). In our implementation, we have different learnable scalars, not only for different Fourier components, but also for different heads and query dimensions.

## 3.3. Multilayer Perceptron

In a transformer architecture, the self-attention layer is followed by an MLP, which consists of two linear layers separated by a non-linearity (Vaswani et al., 2017). The linear layers themselves do not affect equivariance as they just mix together the channels for each irrep:

$$f^{\text{out}}(\mathbf{x}, \rho) = \sigma\left(f^{\text{in}}(\mathbf{x}, \rho)W_1^{(\rho)}\right)W_2^{(\rho)}.$$

Here, $W_1^{(\rho)}, W_2^{(\rho)T} \in \mathbb{C}^{d_{\text{model}} \times d_{\text{hidden}}}$, and $d_{\text{hidden}}$ represents the hidden dimension between the two linear layers. When selecting the non-linearity $\sigma$, it is important to proceed with caution, as commonly used functions like ReLU or sigmoid

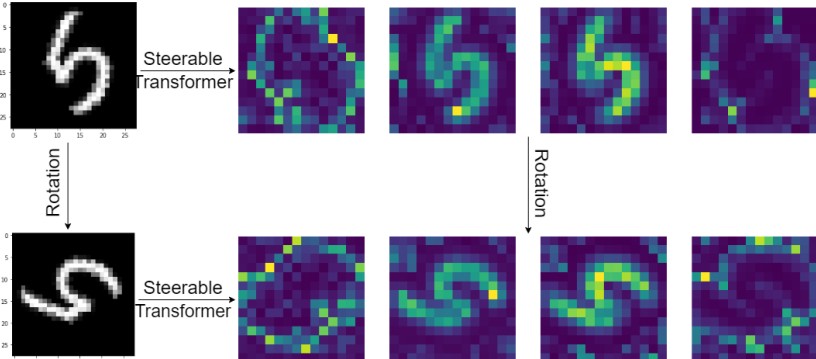

Figure 3: The figure demonstrates the equivariance of attention scores in a trained model. For a fixed pixel $\mathbf{x}_i$, we have plotted the maximum attention score for that pixel ($\max_j \alpha(\mathbf{x}_i, \mathbf{x}_j)$). The different subfigures represent individual heads. The first and last heads appear to capture the object's boundary, while the other two heads focus on the object's body.

may break equivariance, as discussed in Section 2.4. For our experiments, we use a Fourier space non-linearity proposed by Worrall et al. (2017), where they apply ReLU to the magnitude of the equivariant vectors:

$$\sigma(f(\mathbf{x}, \rho)) = \frac{\text{ReLU}(||f(\mathbf{x}, \rho)||_2 + b)}{||f(\mathbf{x}, \rho)||_2} \, f(\mathbf{x}, \rho).$$

Here, $b$ is a learnable bias. Since the norm of the features in Fourier space is invariant to rotations, this non-linearity preserves equivariance. Following the convention in Vaswani et al. (2017), we set $d_{\text{hidden}} = 2d_{\text{model}}$.

### 3.4. Layer Normalization

Another crucial component of the transformer architecture is layer normalization. Notably, the norm of equivariant vectors remains invariant under the Fourier transform, as it preserves the $\mathcal{L}_2$ norm. Leveraging this property, we can derive a steerable normalization as

$$\text{LN}(f)(\mathbf{x}, \rho) = \frac{f(\mathbf{x}, \rho)}{\sqrt{\sum_{\rho} ||f(\mathbf{x}, \rho)||_2^2}} \, .$$

Indeed, the sum over $\rho$ theoretically extends over all irreps of the group, but in practical implementations, it is truncated at particular values, which serves as a hyperparameter that can be tuned. This type of normalization was employed in the two dimensional case by Worrall et al. (2017).

### 3.5. Complexity

The computational complexity of a transformer block mainly stems from its self-attention mechanism and MLP. For a sequence of length $N$ and model dimension $C$, the self-attention mechanism has a complexity of $O(NC^2 + N^2C)$, and the MLP adds $O(NC^2)$, totaling $O(NC^2 + N^2C)$. When $N \asymp C$, this becomes $O(NC^2)$. In steerable self-attention mechanisms, each component effectively has

$d_{\rho}C$ channels, leading to a complexity of $O(Nd_{\rho}^2C^2)$. Analogously, vanilla convolutions in $d$ dimensions, processing $N$ pixels with $C$ input and output channels with a kernel size $k^d$ has a complexity of $O(NC^2k^d)$. In steerable convolutions, replacing $C$ with $d_{\rho}C$ yields $O(Nd_{\rho}^2C^2k^d)$.

In summary, with a fixed kernel size, the computational complexity of steerable self-attention matches that of steerable convolution. Additionally, the increase in complexity from standard to steerable methods is similar for both self-attention mechanisms and convolutions when using many channels and small kernels, as is common in practice. This parallel implies that the additional complexity introduced by steerable methods scales proportionally across both self-attention mechanisms and convolution operations.

## 4. Experiments

We evaluate the performance of our steerable transformer architecture in both two and three dimensions. Specifically, we focus on two vision tasks: image classification and semantic segmentation. In all experiments, we employed a hybrid architecture that integrates steerable convolutions with steerable transformer, and the trained using the Adam Optimizer (Kingma & Ba, 2014).

The steerable convolution encoder is composed of multiple steerable convolution blocks. A steerable convolution block consists of two steerable convolution layers, separated by non-linearity, and is followed by layer normalization. Each steerable convolution block is followed by an average pooling layer. In the baseline models for the classification tasks, this encoder is followed by a final convolution layer with a kernel size matching the input size, effectively serving as a steerable analogue to a flattening layer. In the steerable transformer model, a steerable transformer with two blocks is inserted before this flattening layer. A norm operation is then applied to enforce invariance to both

| Datasets | Model | Frequency Cutoff | Accuracy / Dice Score | Parameter ($\sim \times 10^6$) | Avg Runtime Per Batch | |
|---|---|---|---|---|---|---|
| | | | | | Train (s) | Inference (s) |
| Rotated MNIST | Steerable Convolution | $k = 4$ | $98.72_{\pm 0.02}$ | 1.18 | 0.14 | 0.08 |
| | | $k = 8$ | $98.97_{\pm 0.01}$ | 2.54 | 0.31 | 0.15 |
| | Steerable Transformer | $k = 4$ | $\mathbf{98.82}_{\pm 0.04}$ | 1.13 | 0.13 | 0.08 |
| | | $k = 8$ | $\mathbf{99.03}_{\pm 0.04}$ | 2.24 | 0.30 | 0.14 |
| ModelNet10 ($z$ Rotation) (SO(3) Rotation) | Steerable Convolution | $\ell = 3$ | $90.13_{\pm 0.52}$ $86.62_{\pm 0.25}$ | 1.08 | 1.22 | 0.46 |
| | Steerable Transformer | $\ell = 3$ | $\mathbf{90.40}_{\pm 0.25}$ $\mathbf{86.80}_{\pm 0.58}$ | 0.92 | 1.20 | 0.45 |
| PH2 | Steerable Convolution | $k = 4$ | $89.31_{\pm 0.17}$ | 0.35 | 0.27 | 0.09 |
| | | $k = 8$ | $89.63_{\pm 0.30}$ | 0.96 | 0.67 | 0.19 |
| | Steerable Transformer | $k = 4$ | $\mathbf{90.61}_{\pm 0.90}$ | 0.44 | 0.28 | 0.10 |
| | | $k = 8$ | $\mathbf{90.72}_{\pm 0.70}$ | 1.22 | 0.72 | 0.20 |
| BraTS (Enhancing Tumor) (Tumor Core) (Whole Tumor) | Steerable Convolution | $\ell = 2$ | $73.29_{\pm 0.76}$ $51.07_{\pm 1.23}$ $74.43_{\pm 0.77}$ | 0.12 | 5.04 | 1.00 |
| | Steerable Transformer | $\ell = 2$ | $\mathbf{75.01}_{\pm 0.32}$ $\mathbf{54.89}_{\pm 0.76}$ $\mathbf{76.37}_{\pm 0.46}$ | 0.16 | 5.37 | 1.01 |

Table 1: Comparison of steerable transformers and steerable convolutions. The mean and sd are reported for 5 runs. For ModelNet10 we have reported both the $z$ rotation and SO(3) rotation variations. For PH2 dataset we have reported the dice score for segmentation of the binary mask. For the BraTS dataset we have reported the dice score individually for each tumor category.

rotation and translation. The output is passed through two fully connected layers, with a ReLU activation in between and a dropout layer with a probability of $0.7$ (Marcos et al., 2017). We evaluate performance on two datasets: Rotated MNIST (2D) and ModelNet10 (3D).

For the segmentation tasks, we used a U-net architecture (Ronneberger et al., 2015), with steerable convolution blocks and average pooling to downsample the input image, and a combination of steerable convolution blocks and interpolation to upsample the features back to the input resolution. The model includes skip connections, where features from the downsampling path are added to the corresponding resolution features in the upsampling path. In the baseline model, an MLP is placed between the downsampling and upsampling paths, while in the steerable transformer model, this is replaced by a steerable transformer with two blocks. We report results on two datasets: PH2 (2D) and BraTS (3D).

Table 1 compares the baseline steerable convolutional architecture with the proposed steerable transformer architecture. For classification tasks, we report accuracy, while for segmentation tasks, we present the dice score for each class (excluding the background). The results in Table 1 demonstrate consistent performance improvements with the addition of the steerable transformer encoder layer across various datasets. For comparisons with

other methods, refer to Tables 2 and 3 in Appendix C. Below, we briefly describe the datasets used in our experiments. Further details regarding the architecture and hyperparameters are provided in Appendix C. The code for all experiments is available at `https://github.com/SoumyabrataKundu/Steerable-Transformer`.

**Rotated MNIST:** Rotated MNIST, a variant of the original MNIST dataset (LeCun et al., 2010), includes grayscale images of handwritten digits with resolution $28 \times 28$ that have been randomly rotated. The dataset contains 12,000 training images and 50,000 testing images.

**ModelNet10:** The ModelNet10 dataset (Wu et al., 2015) consists of 3D CAD models from 10 common object categories, with a train/test split of 3991:908. Each category includes triangular meshes representing objects at various orientations and scales. For our experiments, we used the point cloud version of the dataset available at `https://github.com/antao97/PointCloudDatasets`, where the authors employed the farthest point sampling algorithm to sample 2048 points from the surface of each object. These point clouds were then embedded into grids to create voxelized representations. Additionally, rotated versions of the dataset was generated by applying random rotations using nearest-neighbor interpolation (see Figure 5).

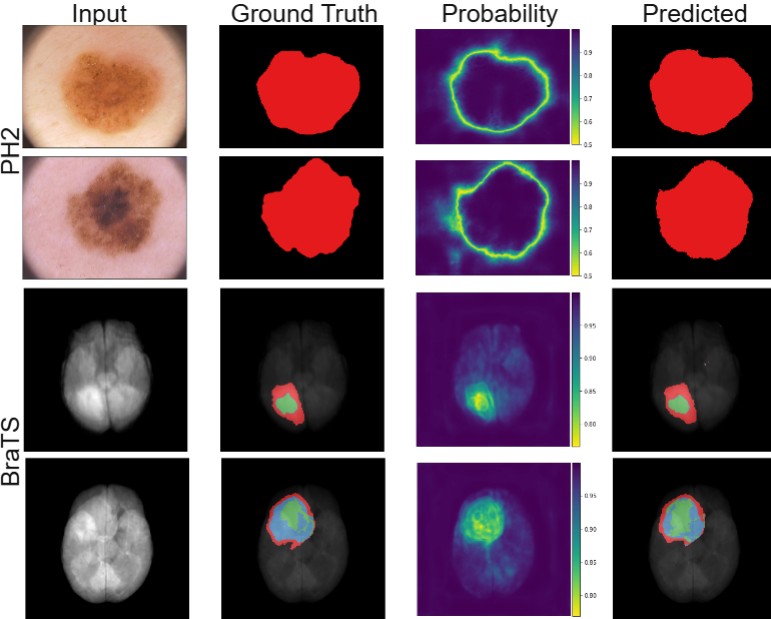

Figure 4: Illustration of ground truth and predicted segmentation, along with the predicted probability for the true class for two examples from each dataset. The red in the PH2 example represent the binary mask. The red, blue and green in the BraTS dataset represent enhancing tumor, tumor core and whole tumor respectively.

**PH2:** The PH2 dataset (Mendonça et al., 2013) is a collection of 200 dermoscopic images used for research in classification and segmentation of skin lesions. We randomly split the dataset into 100:50:50 for training, testing, and validation, respectively. Each image has a resolution of $578\times770$ with RGB channels, and is accompanied by binary masks of the same resolution that delineate lesion regions.

**BraTS:** The Brain Tumor Segmentation (BraTS) dataset (Menze et al., 2015) is used for developing and evaluating algorithms for brain tumor segmentation in MRI scans. It includes pre-operative MRI scans across four modalities, with each image having a resolution of $240\times240\times155$ and annotated to delineate three tumor regions. We used a train/validation/test split of 243:96:145.

## 5. Limitations

While transformer architectures offer powerful modeling capabilities, they also present significant computational challenges. The dot product structure in the attention mechanism can lead to substantial memory consumption, especially when applied to high-resolution images. This limits both the size of the network and the batch size during training, thereby constraining the ability to fully leverage the potential of attention based models. To address this, scaling up the architecture is crucial for unlocking the full benefits of attention in improving model perfor-

mance. In our current experiments, we employed relatively modest Fourier cutoffs, which served as a computational compromise. However, we hypothesize that increasing these cutoff values could lead to better results. Realizing this improvement within a steerable transformer framework would, however, require the model to be scaled up significantly to handle the increased complexity and dimensionality of the representation.

## 6. Conclusion

The transformer architecture has rapidly gained traction across a wide range of machine learning domains, owing to its remarkable capacity to model long range dependencies and contextual relationships within the data. In this work, we present steerable transformers, a novel architectural component designed to work in tandem with steerable convolutions. While convolutions are highly effective at encoding local geometric structures, they are inherently limited in their ability to capture interactions across distant spatial regions. By integrating the transformer's self-attention mechanism, we create a hybrid architecture that combines the strengths of both local and global representations. Our experimental results demonstrate that this integration leads to measurable performance improvements. Furthermore, the steerable transformer framework holds significant promise for high stakes applications like medical imaging, where equivariance to rotations and translations is essential for robust and interpretable analysis.

## Impact Statement

This paper contributes to the ongoing advancement of machine learning, with a particular focus on computer vision and the development of equivariant neural network architectures. By introducing novel techniques that respect the symmetries inherent in visual data, our work pushes the boundaries of how neural networks can effectively and efficiently process complex spatial information. Beyond the technical contributions, this research carries the potential for meaningful societal impact. In particular, the proposed architecture can be applied to biomedical imaging, a field where accurate interpretation of spatial patterns is critical. Enhanced models for tasks such as tumor detection, disease classification, and anatomical structure segmentation could significantly improve the capabilities of AI-assisted diagnosis and precision medicine. By enabling more reliable and interpretable machine learning systems in healthcare, our work supports the broader goal of making medical technologies more effective, accessible, and equitable.

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

# A. Prerequisites

## A.1. Groups

**Definition:** A group $G$ is a non-empty set together with a binary operation (commonly denoted by "·", that combines any two elements $a$ and $b$ of $G$ to form an element of $G$, denoted $a \cdot b$, such that the following three requirements, known as group axioms, are satisfied:

- *Associativity* : For all $a, b, c \in G$, one has $(a \cdot b) \cdot c = a \cdot (b \cdot c)$

- *Identity Element* : There exists as element $e \in G$ such that for every $a \in G$, one has $e \cdot a = a \cdot e = a$. Such an element is unique. It is called the *identity element* of the group.

- *Inverse* : For each $a \in G$, there exists an element $b \in G$ such that $a \cdot b = b \cdot a = e$, where $e$ is the identity element. For each $a$, the element $b$ is unique and it is called the inverse of $a$ and is commonly denoted by $a^{-1}$.

Examples include set of integers $\mathbb{Z}$ with the addition operation and the set of non-zero reals $\mathbb{R} \setminus \{0\}$ with the multiplication operation. From here on we will drop "·", for simplicity. The group operation will be clear from the elements of the group concerned.

**Group Homomorphism:** Given two groups $G$ and $H$, a function $\phi : G \to H$ is called a group homomorphism if $\phi(ab) = \phi(a)\phi(b)$ for any $a, b \in G$. If the map $\phi$ is a bijection, it is called an *isomorphism*. Furthermore, if $G = H$, then an isomorphism is called *automorphism*. The set of all automorphisms of a group $G$ with the operation of composition form a group in itself and is denoted by $\text{Aut}(G)$.

**Compact Groups:** A topological group is a topological space that is also a group such that the group operation and the inverse map are continuous. A compact group is a topological group whose topology realizes it as a compact topological space (see (Munkres, 1974) for definition of compact topological spaces). Some classic examples of compact groups are the groups $\text{SO}(d)$ (the group of all real orthogonal matrices in $d$ dimensions with determinant 1), $\text{U}(d)$ (the group of all complex unitary matrices) and $\text{SU}(d)$ (the group of all complex unitary matrices with determinant 1).

*Special Orthonormal Group* $\text{SO}(d)$*:* This group comprises all real orthogonal matrices in $d$ dimensions with a determinant of 1. These groups are associated with rotation matrices in $d$ dimensions. in two dimensions, the group $\text{SO}(2)$ can be parametrized by a single angle $\theta$ corresponding to the rotation matrix

$$R(\theta) = \begin{bmatrix} \cos(\theta) & \sin(\theta) \\ -\sin(\theta) & \cos(\theta). \end{bmatrix}$$

Here, $R(\theta)$ signifies rotation in the $x$-$y$ plane by an angle $\theta \in [0, 2\pi)$. In three dimensions, the group $\text{SO}(3)$ can be parameterized using the so called *Euler angles*. Consider the rotation matrices $R_z(\alpha)$ and $R_y(\beta)$, defined by

$$R_z(\alpha) = \begin{bmatrix} \cos(\alpha) & -\sin(\alpha) & 0 \\ \sin(\alpha) & \cos(\alpha) & 0 \\ 0 & 0 & 1 \end{bmatrix}, \quad R_y(\beta) = \begin{bmatrix} \cos(\beta) & 0 & \sin(\beta) \\ 0 & 1 & 0 \\ -\sin(\beta) & 0 & \cos(\beta) \end{bmatrix},$$

which represent rotations by angles $\alpha$ and $\beta$ about the $z$-axis and $y$-axis, respectively. Using the $z$-$y$-$z$ convention, any rotation matrix $R \in \text{SO}(3)$ can be expressed as $R = R_z(\alpha)R_y(\beta)R_z(\gamma)$, where $\alpha, \gamma \in [0, 2\pi)$ and $\beta \in [0, \pi)$ are the Euler angles.

**Semi-direct Product Groups:** Given two groups $N$ and $H$ and a group homomorphism $\phi : H \to \text{Aut}(N)$, we can construct a new group $N \rtimes_\phi H$ defined as follows:

- The underlying set is the Cartesian product $N \times H$.

- The group operation is given by $(n_1, h_1)(n_2, h_2) = (n_1 \phi_{h_1}(n_2), h_1 h_2)$.

*Special Euclidean Group SE(d):* The Special Euclidean group $\mathrm{SE}(d)$ consists of all combinations of rotations and translations in $d$ dimensions. Translations in $\mathbb{R}^d$ form a group isomorphic to $\mathbb{R}^d$ itself, while rotations are represented by the group $\mathrm{SO}(d)$. We define a group homomorphism $\phi : \mathrm{SO}(d) \to \mathrm{Aut}(\mathbb{R}^d)$ by setting $\phi_R(\mathbf{t}) = R\mathbf{t}$, which describes how a rotation acts on a translation vector. Using this, the group $\mathrm{SE}(d)$ can be constructed as the semidirect product $\mathbb{R}^d \rtimes_\phi \mathrm{SO}(d)$. The group operation for two elements $(\mathbf{t}_1, R_1)$ and $(\mathbf{t}_2, R_2)$ in $\mathrm{SE}(d)$ is given by

$$(\mathbf{t}_1, R_1)(\mathbf{t}_2, R_2) = (\mathbf{t}_1 + R_1\mathbf{t}_2,\ R_1 R_2).$$

The identity element of $\mathrm{SE}(d)$ is $(\mathbf{0}, I)$, and the inverse of an element $(\mathbf{t}, R)$ is given by $(\mathbf{t}, R)^{-1} = (-R^{-1}\mathbf{t}, R^{-1})$.

### A.2. Group Actions

**Definition:** If $G$ is a group with identity element $e$, and $X$ is a set, then a (left) group action of $G$ on $X$ is a function $\alpha : G \times X \to X$, that satisfies the following two axioms for all $g, h \in G$ and $x \in X$:

- *Identity*: $\alpha(e, x) = x$,

- *Compatibility*: $\alpha(g, \alpha(h, x)) = \alpha(gh, x)$.

Often $\alpha(g, x)$ is shortened to $g \cdot x$. Any group $G$ acts on itself by the group operation. If $G$ acts on $X$, then it also naturally acts on any function $f$ defined on $X$, as $(g \cdot f)(x) = f(g^{-1} \cdot x)$.

*Action of* $\mathrm{SE}(d)$ *on* $\mathbb{R}^d$: The special Euclidean group acts on a vector in $\mathbb{R}^d$ by first applying the rotation component followed by translation. For $\mathbf{x} \in \mathbb{R}^d$ and $(\mathbf{t}, R) \in \mathrm{SE}(d)$,

$$(\mathbf{t}, R) \cdot \mathbf{x} = R\mathbf{x} + \mathbf{t}$$

gives us the action of $\mathrm{SE}(d)$ on $\mathbb{R}^d$.

### A.3. Group Representations

**Definition:** A representation of a group $G$ is a group homomorphism from $G$ to $\mathrm{GL}(\mathbb{C}^n)$ (group of invertible linear maps on $\mathbb{C}^n$). Here $n$ is called the dimension of the representation, which can possibly be infinite. A representation is *unitary* if $\rho$ maps to unitary linear transformation of $\mathbb{C}^n$.

**Irreducible Representations:** If we have two representations, $\rho_1$ and $\rho_2$ of dimensions $n_1$ and $n_2$ respectively, then the two can be combined by a direct sum to give another representation of dimension $n_1 + n_2$,

$$\rho_1(g) \oplus \rho_2(g) = \begin{bmatrix} \rho_1(g) & 0 \\ 0 & \rho_2(g) \end{bmatrix}.$$

A representation is said to be *completely reducible* if it can be expressed as a direct sum of other representations after maybe a change of basis, i.e,

$$U\rho(g)U^{-1} = \bigoplus_i \rho_i(g)$$

where $U$ is a unitary change of basis matrix and the direct sum extends over some number of representations. However, for *every* group there are a some representations which cannot be broken further into a direct sum of other representations. These are called the *irreducible representations* or *irreps* of the group. These irreps are the building blocks of the all other representations of the group, in the sense that any representation can be written as a direct sum of the irreps:

$$\rho(g) = U \left[ \bigoplus_i \rho^{(i)}(g) \right] U^{-1},$$

where again $U$ is a change of basis matrix and $\rho^{(i)}$ are the irreps. The Peter–Weyl Theorem by (Peter & Weyl, 1927) tells us that for a compact group $G$, any unitary representation $\rho$ is completely reducible and splits into direct sum of irreducible *finite dimensional unitary* representations of $G$.

**Irreducible Representations of** $\mathrm{SO}(2)$ **and** $\mathrm{SO}(3)$: $\mathrm{SO}(d)$ being a compact group, all its irreps are finite dimensional unitary representations. In the case of $\mathrm{SO}(2)$, every irrep is one-dimensional and indexed by an integer. The group $\mathrm{SO}(2)$ can be parameterized by an angle $\theta \in [0, 2\pi)$, corresponding to a rotation in the $x$-$y$ plane by $\theta$ radians. Under this parameterization, the irreps of $\mathrm{SO}(2)$ take the form

$$\rho^{(k)}(\theta) = e^{ik\theta}, \qquad k \in \mathbb{Z}.$$

The irreps of $\mathrm{SO}(3)$ are indexed by positive integers $\ell \in \mathbb{Z}_{\geq 0}$, where the $\ell$'th representation is of dimension $2\ell + 1$ and are given by the so called Wigner D-matrices (Wigner, 1932):

$$\rho^{(\ell)}(R) = D^{(\ell)}(R), \qquad \ell \in \mathbb{Z}_{\geq 0}.$$

### A.4. Fourier Transform

**Haar Measure:** There is, up to a positive multiplicative constant, a unique countably additive, nontrivial measure $\mu$ on the Borel subsets of $G$ satisfying the following properties:

- $\mu$ is left-translation-invariant: $\mu(gS) = \mu(S)$ for every $g \in G$ and all Borel sets $S \subseteq G$.

- $\mu$ is finite on every compact set: $\mu(K) < \infty$ for all compact $K \subseteq G$.

- $\mu$ is outer regular on Borel sets $S \subseteq G$: $\mu(S) = \inf\{\mu(U) : S \subseteq U, U \text{ open}\}$.

- $\mu$ is inner regular on open sets $U \subseteq G$: $\mu(U) = \sup\{\mu(K) : K \subseteq U, K \text{ compact}\}$.

**Fourier Transform on Compact groups:** A notable and useful property of compact groups is that the set of (isomorphism classes of) their irreps is countable (Robert, 1983). This fact underpins a powerful generalization of classical Fourier analysis to the setting of compact groups, enabling square-integrable functions to be decomposed into frequency components indexed by group representations. Let $f \in \mathcal{L}_2(G)$, where $\mathcal{L}_2(G)$ denotes the space of complex-valued, square-integrable functions on $G$ with respect to the (normalized) Haar measure. For each irrep $\rho$, the Fourier transform of $f$ at $\rho$ is defined as

$$\widehat{f}(\rho) = \int_G f(g)\, \rho(g)\, d\mu(g),$$

where $\widehat{f}(\rho)$ is a complex matrix of size $d_\rho \times d_\rho$. The inverse Fourier transform allows one to reconstruct the original function $f$ from its Fourier coefficients via the formula

$$f(g) = \sum_\rho d_\rho \operatorname{Tr}\left(\widehat{f}(\rho)\, \rho(g)^\dagger\right),$$

where the convergence is in the $\mathcal{L}_2$-sense. This formula is justified by the Peter—Weyl theorem, which states that the matrix coefficients of all irreps form a complete orthonormal basis for $\mathcal{L}_2(G)$. Consequently, any square integrable function on $G$ can be expressed as a (generalized) Fourier series in terms of these basis functions.

## B. Proof of Theorem 1

*Proof.* Throughout the proof we will assume a single input channel ($d_{\text{model}} = 1$). Fix $(\mathbf{t}, R) \in \text{SE}(d)$. By equation (6), under the action of $(\mathbf{t}, R)^{-1}$, any input $f^{\text{in}}$ transforms as

$$f^{\text{in}}(\mathbf{x}) \mapsto [(\mathbf{t}, R)^{-1} \cdot f^{\text{in}}](\mathbf{x}) = \rho(R)^\dagger f^{\text{in}}(R\mathbf{x} + \mathbf{t}).$$

Analogously, the query, key and value vectors in equations (10)-(12) transform as

$$\mathbf{q}^{(\rho)}(\mathbf{x}) \mapsto [(\mathbf{t}, R)^{-1} \cdot f^{\text{in}}](\mathbf{x})\mathbf{W}_Q^{(\rho)} = \rho(R)^\dagger f^{\text{in}}(R\mathbf{x} + \mathbf{t})\mathbf{W}_Q^{(\rho)} = \rho(R)^\dagger \mathbf{q}^{(\rho)}(R\mathbf{x} + \mathbf{t}) \tag{20}$$

$$\mathbf{k}^{(\rho)}(\mathbf{x}, \mathbf{y}) \mapsto [(\mathbf{t}, R)^{-1} \cdot f^{\text{in}}](\mathbf{x})\mathbf{W}_K^{(\rho)} + \mathbf{P}^{(\rho)}(\mathbf{x}, \mathbf{y}) = \rho(R)^\dagger f^{\text{in}}(R\mathbf{x} + \mathbf{t})\mathbf{W}_K^{(\rho)} + \mathbf{P}^{(\rho)}(\mathbf{x}, \mathbf{y}) \tag{21}$$

$$\mathbf{v}^{(\rho)}(\mathbf{x}, \mathbf{y}) \mapsto [(\mathbf{t}, R)^{-1} \cdot f^{\text{in}}](\mathbf{x})\mathbf{W}_V^{(\rho)} + \mathbf{P}^{(\rho)}(\mathbf{x}, \mathbf{y}) = \rho(R)^\dagger f^{\text{in}}(R\mathbf{x} + \mathbf{t})\mathbf{W}_V^{(\rho)} + \mathbf{P}^{(\rho)}(\mathbf{x}, \mathbf{y}), \tag{22}$$

respectively. The score function $s$ is computed using a scaled dot product. Using the fact that $\rho$ is unitary, the score function transforms as

$$s(\mathbf{x}, \mathbf{y})$$

$$= \sum_\rho d_K^{-1/2} \text{vec}\left(\mathbf{q}^{(\rho)}(\mathbf{x})\right)^\dagger \text{vec}\left(\mathbf{k}^{(\rho)}(\mathbf{x}, \mathbf{y})\right)$$

*(by equation (13))*

$$\mapsto \sum_\rho d_K^{-1/2} \text{vec}\left(\rho(R)^\dagger \mathbf{q}^{(\rho)}(R\mathbf{x} + \mathbf{t})\right)^\dagger \text{vec}\left(\rho(R)^\dagger f^{\text{in}}(R\mathbf{x} + \mathbf{t})\mathbf{W}_K^{(\rho)} + \mathbf{P}^{(\rho)}(\mathbf{x}, \mathbf{y})\right)$$

*(by equation (20)-(21))*

$$= \sum_\rho d_K^{-1/2} \left(\rho(R)^\dagger \mathbf{q}^{(\rho)}(R\mathbf{x} + \mathbf{t})\right)^\dagger \left(\rho(R)^\dagger f^{\text{in}}(R\mathbf{x} + \mathbf{t})\mathbf{W}_K^{(\rho)} + \mathbf{P}^{(\rho)}(\mathbf{x}, \mathbf{y})\right)$$

*(since we assumed $d_{\text{model}} = 1$)*

$$= \sum_\rho d_K^{-1/2} \mathbf{q}^{(\rho)}(R\mathbf{x} + \mathbf{t})^\dagger \rho(R)\rho(R)^\dagger f^{\text{in}}(R\mathbf{x} + \mathbf{t})\mathbf{W}_K^{(\rho)} + d_K^{-1/2} \mathbf{q}^{(\rho)}(R\mathbf{x} + \mathbf{t})^\dagger \rho(R)\mathbf{P}^{(\rho)}(\mathbf{x}, \mathbf{y})$$

$$= \sum_\rho d_K^{-1/2} \mathbf{q}^{(\rho)}(R\mathbf{x} + \mathbf{t})^\dagger f^{\text{in}}(R\mathbf{x} + \mathbf{t})\mathbf{W}_K^{(\rho)} + d_K^{-1/2} \mathbf{q}^{(\rho)}(R\mathbf{x} + \mathbf{t})^\dagger \rho(R)\mathbf{P}^{(\rho)}(\mathbf{x}, \mathbf{y})$$

*(since $\rho(R)$ is unitary)*

$$= \sum_\rho d_K^{-1/2} \mathbf{q}^{(\rho)}(R\mathbf{x} + \mathbf{t})^\dagger \left(f^{\text{in}}(R\mathbf{x} + \mathbf{t})\mathbf{W}_K^{(\rho)} + \mathbf{P}^{(\rho)}(R\mathbf{x} + \mathbf{t}, R\mathbf{y} + \mathbf{t})\right)$$

$$+ \sum_\rho d_K^{-1/2} \mathbf{q}^{(\rho)}(R\mathbf{x} + \mathbf{t})^\dagger \left[\rho(R)\mathbf{P}^{(\rho)}(\mathbf{x}, \mathbf{y}) - \mathbf{P}^{(\rho)}(R\mathbf{x} + \mathbf{t}, R\mathbf{y} + \mathbf{t})\right]$$

$$= \sum_\rho d_K^{-1/2} \mathbf{q}^{(\rho)}(R\mathbf{x} + \mathbf{t})^\dagger \mathbf{k}^{(\rho)}(R\mathbf{x} + \mathbf{t}, R\mathbf{y} + \mathbf{t}) + \sum_\rho d_K^{-1/2} \mathbf{q}^{(\rho)}(R\mathbf{x} + \mathbf{t})^\dagger \Delta^{(\rho)}(R\mathbf{x} + \mathbf{t}, R\mathbf{y} + \mathbf{t})$$

*(by equation (11))*

$$= \sum_\rho d_K^{-1/2} \text{vec}\left(\mathbf{q}^{(\rho)}(R\mathbf{x} + \mathbf{t})\right)^\dagger \text{vec}\left(\mathbf{k}^{(\rho)}(R\mathbf{x} + \mathbf{t}, R\mathbf{y} + \mathbf{t})\right) + \sum_\rho d_K^{-1/2} \mathbf{q}^{(\rho)}(R\mathbf{x} + \mathbf{t})^\dagger \Delta^{(\rho)}(R\mathbf{x} + \mathbf{t}, R\mathbf{y} + \mathbf{t})$$

*(since we assumed $d_{\text{model}} = 1$)*

$$= s(R\mathbf{x} + \mathbf{t}, R\mathbf{y} + \mathbf{t}) + \sum_\rho d_K^{-1/2} \mathbf{q}^{(\rho)}(R\mathbf{x} + \mathbf{t})^\dagger \Delta^{(\rho)}(R\mathbf{x} + \mathbf{t}, R\mathbf{y} + \mathbf{t}), \tag{23}$$

*(by equation (13))*

where $\Delta^{(\rho)}(R\mathbf{x} + \mathbf{t}, R\mathbf{y} + \mathbf{t}) := \rho(R)\mathbf{P}^{(\rho)}(\mathbf{x}, \mathbf{y}) - \mathbf{P}^{(\rho)}(R\mathbf{x} + \mathbf{t}, R\mathbf{y} + \mathbf{t})$. The attention scores are obtained by taking the softmax of the scores (23). Therefore, the attention scores transform as

$$\alpha(\mathbf{x}, \mathbf{y})$$

$$= \frac{\exp\left(|s(\mathbf{x}, \mathbf{y})|\right)}{\int_{\Omega(f^{\text{in}})} \exp\left(|s(\mathbf{x}, \mathbf{z})|\right) \, d\mathbf{z}}$$

(*by equation* (14))

$$\mapsto \frac{\exp\left(\left|s(R\mathbf{x}+\mathbf{t}, R\mathbf{y}+\mathbf{t}) + \sum_\rho d_K^{-1/2} \mathbf{q}^{(\rho)}(R\mathbf{x}+\mathbf{t})^\dagger \Delta^{(\rho)}(R\mathbf{x}+\mathbf{t}, R\mathbf{y}+\mathbf{t})\right|\right)}{\int_{\Omega((\mathbf{t},R)^{-1}\cdot f^{\text{in}})} \exp\left(\left|s(R\mathbf{x}+\mathbf{t}, R\mathbf{z}+\mathbf{t}) + \sum_\rho d_K^{-1/2} \mathbf{q}^{(\rho)}(R\mathbf{x}+\mathbf{t})^\dagger \Delta^{(\rho)}(R\mathbf{x}+\mathbf{t}, R\mathbf{z}+\mathbf{t})\right|\right) \, d\mathbf{z}}$$

(*by equation* (23))

$$= \frac{\exp\left(\left|s(R\mathbf{x}+\mathbf{t}, R\mathbf{y}+\mathbf{t}) + \sum_\rho d_K^{-1/2} \mathbf{q}^{(\rho)}(R\mathbf{x}+\mathbf{t})^\dagger \Delta^{(\rho)}(R\mathbf{x}+\mathbf{t}, R\mathbf{y}+\mathbf{t})\right|\right)}{\int_{(\mathbf{t},R)^{-1}\cdot\Omega(f^{\text{in}})} \exp\left(\left|s(R\mathbf{x}+\mathbf{t}, R\mathbf{z}+\mathbf{t}) + \sum_\rho d_K^{-1/2} \mathbf{q}^{(\rho)}(R\mathbf{x}+\mathbf{t})^\dagger \Delta^{(\rho)}(R\mathbf{x}+\mathbf{t}, R\mathbf{z}+\mathbf{t})\right|\right) \, d\mathbf{z}}. \tag{24}$$

(*if* $\Omega(f^{\text{in}})$ *is the support of* $f^{\text{in}}$, *then* $(\mathbf{t}, R)^{-1} \cdot \Omega(f^{\text{in}})$ *is the support of* $(\mathbf{t}, R)^{-1} \cdot f^{\text{in}}$)

Define the expression in (24) as $\beta(R\mathbf{x}+\mathbf{t}, R\mathbf{y}+\mathbf{t})$. Using these attention scores, the output of the attention mechanism $f^{\text{out}}$, for any input $f^{\text{in}}$, transforms as

$$f^{\text{out}}(\mathbf{x})$$
$$= \int_{\Omega(f^{\text{in}})} \alpha(\mathbf{x}, \mathbf{y}) \mathbf{v}^{(\rho)}(\mathbf{x}, \mathbf{y}) \, d\mathbf{y}$$

(*by equation* (15))

$$\mapsto \int_{(\mathbf{t},R)^{-1}\cdot\Omega(f^{\text{in}})} \beta(R\mathbf{x}+\mathbf{t}, R\mathbf{y}+\mathbf{t}) \left[\rho(R)^\dagger f^{\text{in}}(R\mathbf{x}+\mathbf{t})\mathbf{W}_V^{(\rho)} + \mathbf{P}^{(\rho)}(\mathbf{x}, \mathbf{y})\right] d\mathbf{y}$$

(*by equation* (22))

$$= \rho(R)^\dagger \int_{(\mathbf{t},R)^{-1}\cdot\Omega(f^{\text{in}})} \beta(R\mathbf{x}+\mathbf{t}, R\mathbf{y}+\mathbf{t}) \left[f^{\text{in}}(R\mathbf{x}+\mathbf{t})\mathbf{W}_V^{(\rho)} + \mathbf{P}^{(\rho)}(R\mathbf{x}+\mathbf{t}, R\mathbf{y}+\mathbf{t})\right] d\mathbf{y}$$
$$+ \int_{(\mathbf{t},R)^{-1}\cdot\Omega(f^{\text{in}})} \beta(R\mathbf{x}+\mathbf{t}, R\mathbf{y}+\mathbf{t}) \left[\mathbf{P}^{(\rho)}(\mathbf{x}, \mathbf{y}) - \rho(R)^\dagger \mathbf{P}^{(\rho)}(R\mathbf{x}+\mathbf{t}, R\mathbf{y}+\mathbf{t})\right] d\mathbf{y}$$
$$= \rho(R)^\dagger \int_{(\mathbf{t},R)^{-1}\cdot\Omega(f^{\text{in}})} \beta(R\mathbf{x}+\mathbf{t}, R\mathbf{y}+\mathbf{t}) \mathbf{v}^{(\rho)}(R\mathbf{x}+\mathbf{t}, R\mathbf{y}+\mathbf{t}) \, d\mathbf{y}$$
$$+ \rho(R)^\dagger \int_{(\mathbf{t},R)^{-1}\cdot\Omega(f^{\text{in}})} \beta(R\mathbf{x}+\mathbf{t}, R\mathbf{y}+\mathbf{t}) \Delta^{(\rho)}(R\mathbf{x}+\mathbf{t}, R\mathbf{y}+\mathbf{t}) \, d\mathbf{y}. \tag{25}$$

(*by equation* (12) *and definition of* $\Delta^{(\rho)}$)

According to the statement of the proposition, the output $f^{\text{out}}$ should transform as

$$f^{\text{out}}(\mathbf{x}, \rho) \mapsto \rho(R)^\dagger f^{\text{out}}(R\mathbf{x}+\mathbf{t}, \rho) = \rho(R)^\dagger \int_{\Omega(f^{\text{in}})} \alpha(R\mathbf{x}+\mathbf{t}, \mathbf{y}) \mathbf{v}^{(\rho)}(R\mathbf{x}+\mathbf{t}, \mathbf{y}) \, d\mathbf{y}$$
$$= \rho(R)^\dagger \int_{(\mathbf{t},R)^{-1}\cdot\Omega(f^{\text{in}})} \alpha(R\mathbf{x}+\mathbf{t}, R\mathbf{y}+\mathbf{t}) \mathbf{v}^{(\rho)}(R\mathbf{x}+\mathbf{t}, R\mathbf{y}+\mathbf{t}) \, d\mathbf{y}. \tag{26}$$

The transformations in (25) and (26) are equivalent iff

$$\int_{(\mathbf{t},R)^{-1}\cdot\Omega(f^{\text{in}})} \beta(R\mathbf{x}+\mathbf{t}, R\mathbf{y}+\mathbf{t}) \mathbf{v}^{(\rho)}(R\mathbf{x}+\mathbf{t}, R\mathbf{y}+\mathbf{t}) \, d\mathbf{y} + \int_{(\mathbf{t},R)^{-1}\cdot\Omega(f^{\text{in}})} \beta(R\mathbf{x}+\mathbf{t}, R\mathbf{y}+\mathbf{t}) \Delta^{(\rho)}(R\mathbf{x}+\mathbf{t}, R\mathbf{y}+\mathbf{t}) \, d\mathbf{y}$$
$$= \int_{(\mathbf{t},R)^{-1}\cdot\Omega(f^{\text{in}})} \alpha(R\mathbf{x}+\mathbf{t}, R\mathbf{y}+\mathbf{t}) \mathbf{v}^{(\rho)}(R\mathbf{x}+\mathbf{t}, R\mathbf{y}+\mathbf{t}) \, d\mathbf{y} \tag{27}$$

for all inputs $f^{\text{in}}$, and for all weights $W_Q^{(\rho)}, W_K^{(\rho)}, W_V^{(\rho)}$ for all irreps $\rho$. Note that, if $\Delta^{(\rho)} \equiv 0$ for all irreps $\rho$, then $\alpha \equiv \beta$ by equation (24) and hence, the equality in (27) holds for all $f^{\text{in}}, W_Q^{(\rho)}, W_K^{(\rho)}, W_V^{(\rho)}$. Conversely, suppose the equality (27)

holds for any $f^{\text{in}}, W_Q^{(\rho)}, W_K^{(\rho)}, W_V^{(\rho)}$. In particular, setting $W_Q^{(\rho)} = 0$, by equations (20) and (24), we get $\alpha \equiv \beta \propto 1$, and therefore equation (27) implies

$$\int_{(\mathbf{t},R)^{-1}\cdot\Omega(f^{\text{in}})} \alpha(R\mathbf{x}+\mathbf{t}, R\mathbf{y}+\mathbf{t})\mathbf{v}^{(\rho)}(R\mathbf{x}+\mathbf{t}, R\mathbf{y}+\mathbf{t})\,d\mathbf{y} + \int_{(\mathbf{t},R)^{-1}\cdot\Omega(f^{\text{in}})} \alpha(R\mathbf{x}+\mathbf{t}, R\mathbf{y}+\mathbf{t})\Delta^{(\rho)}(R\mathbf{x}+\mathbf{t}, R\mathbf{y}+\mathbf{t})\,d\mathbf{y}$$

$$= \int_{(\mathbf{t},R)^{-1}\cdot\Omega(f^{\text{in}})} \alpha(R\mathbf{x}+\mathbf{t}, R\mathbf{y}+\mathbf{t})\mathbf{v}^{(\rho)}(R\mathbf{x}+\mathbf{t}, R\mathbf{y}+\mathbf{t})\,d\mathbf{y}$$

$$\implies \int_{(\mathbf{t},R)^{-1}\cdot\Omega(f^{\text{in}})} \alpha(R\mathbf{x}+\mathbf{t}, R\mathbf{y}+\mathbf{t})\Delta^{(\rho)}(R\mathbf{x}+\mathbf{t}, R\mathbf{y}+\mathbf{t})\,d\mathbf{y} = 0$$

$$\implies \int_{(\mathbf{t},R)^{-1}\cdot\Omega(f^{\text{in}})} \Delta^{(\rho)}(R\mathbf{x}+\mathbf{t}, R\mathbf{y}+\mathbf{t})\,d\mathbf{y} = 0. \tag{28}$$

$$(\textit{since } \alpha \propto 1)$$

The equality in (28) holds for any compactly supported $f^{\text{in}}$. In particular, consider a class of functions supported on a ball of radius $1/n$ around $R\mathbf{y}_0 + \mathbf{t}$, with $n \in \mathbb{N}$ and a fixed $\mathbf{y}_0 \in \mathbb{R}^d$, denoted by $B_{1/n}(R\mathbf{y}_0 + \mathbf{t})$. Then, setting $\Omega(f^{\text{in}}) = B_{1/n}(R\mathbf{y}_0 + \mathbf{t})$ in (28), we have

$$\int_{(\mathbf{t},R)^{-1}\cdot B_{1/n}(R\mathbf{y}_0+\mathbf{t})} \Delta^{(\rho)}(R\mathbf{x}+\mathbf{t}, R\mathbf{y}+\mathbf{t})\,d\mathbf{y} = 0 \qquad\qquad \text{for all } n \in \mathbb{N}$$

$$\implies \int_{B_{1/n}(\mathbf{y}_0)} \Delta^{(\rho)}(R\mathbf{x}+\mathbf{t}, R\mathbf{y}+\mathbf{t})\,d\mathbf{y} = 0 \qquad\qquad \text{for all } n \in \mathbb{N}$$

$$\implies \frac{1}{|B_{1/n}(\mathbf{y}_0)|}\int_{B_{1/n}(\mathbf{y}_0)} \Delta^{(\rho)}(R\mathbf{x}+\mathbf{t}, R\mathbf{y}+\mathbf{t})\,d\mathbf{y} = 0 \qquad\qquad \text{for all } n \in \mathbb{N}$$

$$\implies \lim_{n\to\infty} \frac{1}{|B_{1/n}(\mathbf{y}_0)|}\int_{B_{1/n}(\mathbf{y}_0)} \Delta^{(\rho)}(R\mathbf{x}+\mathbf{t}, R\mathbf{y}+\mathbf{t})\,d\mathbf{y} = 0$$

$$\implies \Delta^{(\rho)}(R\mathbf{x}+\mathbf{t}, R\mathbf{y}_0+\mathbf{t}) = 0. \tag{29}$$

$$(\textit{using continuity of } \mathbf{P}^{(\rho)} \textit{ and Lemma } 1)$$

Since $\mathbf{y}_0 \in \mathbb{R}^d$ was arbitrarily fixed, the equality in (29) hold for all $\mathbf{y}_0 \in \mathbb{R}^d$. Hence, the equality in (27) holds iff

$$\mathbf{P}^{(\rho)}(R\mathbf{x}+\mathbf{t}, R\mathbf{y}+\mathbf{t}) = \rho(R)\mathbf{P}^{(\rho)}(\mathbf{x},\mathbf{y}).$$

for all $\mathbf{x}, \mathbf{y} \in \mathbb{R}^d$ and irreps $\rho$. Since $(\mathbf{t}, R) \in \text{SE}(d)$ was chosen arbitrarily, the result holds for all $(\mathbf{t}, R) \in \text{SE}(d)$, which completes the proof.

$$\square$$

**Lemma 1.** *Suppose $f : \mathbb{R}^d \to \mathbb{C}^p$ is a function continuous at $\mathbf{x}_0 \in \mathbb{R}^d$. Let $B_r(\mathbf{x}_0)$, defined as*

$$B_r(\mathbf{x}_0) := \{\mathbf{x} \in \mathbb{R}^d : \|\mathbf{x}_0 - \mathbf{x}\|_2 \le r\},$$

*denote a ball of radius $r$ around $\mathbf{x}_0$. Then,*

$$\lim_{r\to 0} \frac{1}{|B_r(\mathbf{x}_0)|}\int_{B_r(\mathbf{x}_0)} f(\mathbf{x})\,d\mathbf{x} = f(\mathbf{x}_0)$$

*where $|B_r(\mathbf{x}_0)|$ is the Lebesgue measure of $B_r(\mathbf{x}_0)$.*

*Proof.* By continuity of $f$ at $\mathbf{x}_0$, for $\varepsilon > 0$, there exists $\delta > 0$ such that, for any $0 \le r \le \delta$

$$\|f(\mathbf{x}_0) - f(\mathbf{x})\|_2 \le \varepsilon \tag{30}$$

for all $\mathbf{x} \in B_r(\mathbf{x}_0)$. Therefore, we have

$$\left\|\frac{1}{|B_r(\mathbf{x}_0)|}\int_{B_r(\mathbf{x}_0)} f(\mathbf{x})\,d\mathbf{x} - f(\mathbf{x}_0)\right\|_2 = \left\|\frac{1}{|B_r(\mathbf{x}_0)|}\int_{B_r(\mathbf{x}_0)} f(\mathbf{x}) - f(\mathbf{x}_0)\,d\mathbf{x}\right\|_2$$

$$\leq \frac{1}{|B_r(\mathbf{x}_0)|} \int_{B_r(\mathbf{x}_0)} ||f(\mathbf{x}) - f(\mathbf{x}_0)||_2 \, d\mathbf{x}$$

$$\leq \frac{1}{|B_r(\mathbf{x}_0)|} \int_{B_r(\mathbf{x}_0)} \varepsilon \, d\mathbf{x}$$

(*by equation* (30))

$$= \varepsilon. \tag{31}$$

The inequality in (31) holds for $0 \leq r \leq \delta$. Hence,

$$\lim_{r \to 0} \left\| \frac{1}{|B_r(\mathbf{x}_0)|} \int_{B_r(\mathbf{x}_0)} f(\mathbf{x}) \, d\mathbf{x} - f(\mathbf{x}_0) \right\|_2 = 0.$$

$\square$

## C. Experiment Details

In this section, we provide additional details on the datasets, architectural variations, and hyperparameters for all experiments conducted in this work.

### C.1. Rotated MNIST

| Type | Method | Error% | Parameters |
|---|---|---|---|
| Miscellaneous | SVM (Larochelle et al., 2007) | 11.11 | – |
| | TIRBM (Sohn & Lee, 2012) | 4.2 | – |
| | TI-Pooling (Laptev et al., 2016) | 1.2 | – |
| Equivariant Convolution | P4CNN (Cohen & Welling, 2016) | 2.28 | 25k |
| | Harmonic Net (Worrall et al., 2017) | 1.69 | 33k |
| | RotEqNet (Marcos et al., 2017) | 1.09 | – |
| | Weiler et al. (Weiler et al., 2018b) | 0.71 | 6M |
| | E2CNN (Weiler & Cesa, 2019) | **0.69** | 6M |
| Equivariant Attention | $\alpha$-R4 CNN (Romero et al., 2020) | 1.69 | 73k |
| | GSA-Nets (Romero & Cordonnier, 2021) | 2.03 | 44k |
| | GE-ViT (Xu et al., 2023) | 1.99 | 44k |
| | Our ($k = 4$) | 1.18 | 1.13M |
| | Our ($k = 8$) | 0.97 | 2.24M |

Table 2: Comparison of the performance of the equivariant attentive architecture on the Rotated MNIST dataset.

The network architecture in this experiment consists of three convolutional blocks with progressively increasing numbers of channels. These blocks are separated by average pooling layers, which down-sample the feature maps to improve computational efficiency. The output is flattened and passed through fully connected layers that apply linear transformations, batch normalization, ReLU activation, and dropout to prevent overfitting. The final layer maps the output to ten classes. In the case of the steerable transformer, a transformer block with a single layer follows the convolutional blocks.

The networks were trained using the Adam optimizer (Kingma & Ba, 2014), starting with a learning rate of $5 \times 10^{-3}$, which was reduced by a factor of 0.5 every 20 epochs, along with a weight decay of $5 \times 10^{-4}$ for 150 epochs. A batch size of 25 was used, and training the largest model took 4 hours on a 16GB GPU.

Table 2 compares the performance of the steerable transformers with other attention-based methods reported for this dataset. Our approach significantly outperforms the other attention-based methods, which are standalone, while ours is built on top of convolutions. The only methods that surpass our results, by Weiler et al. (2018b) and Weiler & Cesa (2019), use a Fourier cutoff of 16, while we use a cutoff of 8. We believe this accounts for the performance difference, which is due to implementation and resource limitations, as higher Fourier cutoffs lead to out-of-memory issues.

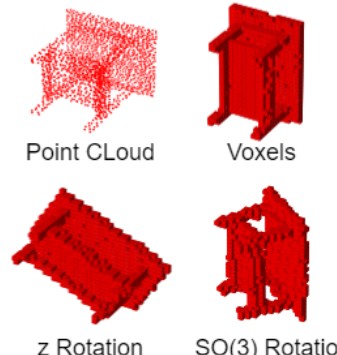

Figure 5: Examples from the ModelNet10 dataset are shown in various formats: point cloud, voxel representation, and rotated perturbations of voxels.

| | Method | Accuracy | Parameters ($\sim \times 10^6$) |
|---|---|---|---|
| Point Cloud | ECC (Simonovsky & Komodakis, 2017) | 90.8 | - |
| | SO-Net (Li et al., 2018) | 93.9 | 2.5 |
| | Rot-SO-Net (Li et al., 2019) | 94.5 | 2.5 |
| Voxel | 3D ShapeNets (Wu et al., 2015) | 83.5 | 12 |
| | VRN (Brock et al., 2016) | 91.3 | 18 |
| | VoxNet (Maturana & Scherer, 2015) | 92.0 | 0.92 |
| | FusionNet (Hegde & Zadeh, 2016) | 93.1 | 120 |
| | ORION (Sedaghat et al., 2016) | 93.8 | 0.91 |
| | Cubenet (Worrall & Brostow, 2018) | **94.6** | 4.5 |
| | Our | 91.1 | 0.92 |

Table 3: Comparison of performance on ModelNet10 with $z$ rotation perturbation. Other methods used both train and test time augmentation, while we applied augmentation only during test time, not during training.

## C.2. ModelNet10

The point cloud data with 2048 points is available at `https://github.com/antao97/PointCloudDatasets`. Similar to the Rotated MNIST experiment, we utilized three convolutional blocks. The features for each irrep are in a $1:1:1:1$ ratio. The number of features per irrep increases with each convolutional block, and the data is downsampled using average pooling. For the steerable transformer, this convolutional encoder is supplemented with a transformer block containing a single transformer layer. After an additional normalization step, the output is flattened and passed through fully connected layers that reduce the dimensionality to 128 features. To ensure training stability and prevent overfitting, batch normalization, ReLU activation, and dropout are applied, with the final output layer classifying the data into ten classes.

The networks were trained using the Adam optimizer (Kingma & Ba, 2014), with an initial learning rate of $1 \times 10^{-3}$, which decreased by a factor of 0.5 every 20 epochs, for a total of 50 epochs. A batch size of 5 was used, and training the largest model took 12 hours on a 16GB GPU.

Table 3 compares our method to other non-attention-based methods on the $z$-rotated version of the dataset. Unlike our approach, which does not use augmentation during training, all other methods were trained with augmentation involving 12 uniformly stratified rotations along the $z$-axis. To ensure a fair comparison, we applied test-time augmentation by averaging the accuracy of all 12 predictions for each test data point. Despite not using train-time augmentation and having fewer parameters, our method achieves performance comparable to these other methods.

## C.3. PH2

The data for the experiment is available at `https://www.fc.up.pt/addi/ph2%20database.html`. The network begins with two convolutional blocks, each composed of steerable convolutional layers, non-linearities, and batch normalization, which progressively downsample the input and extract relevant features. Then, a transformer encoder is used to capture global dependencies in the data. After this, the model uses two convolutional blocks to upsample and refine the extracted features. In between each block, the features are upsampled using bilinear interpolation. Finally, the features are restored to the original resolutions. In the final step, the vector norm is computed, converting the complex-valued feature maps into real values to produce the logits of each class.

The networks were trained using the Adam optimizer (Kingma & Ba, 2014), starting with an initial learning rate of $1 \times 10^{-2}$, which decreased by a factor of 0.5 every 20 epochs, for a total of 100 epochs. We used a batch size of 1, and training the largest model took 2 hours on a 16GB GPU. A larger batch size could not be used due to out-of-memory errors.

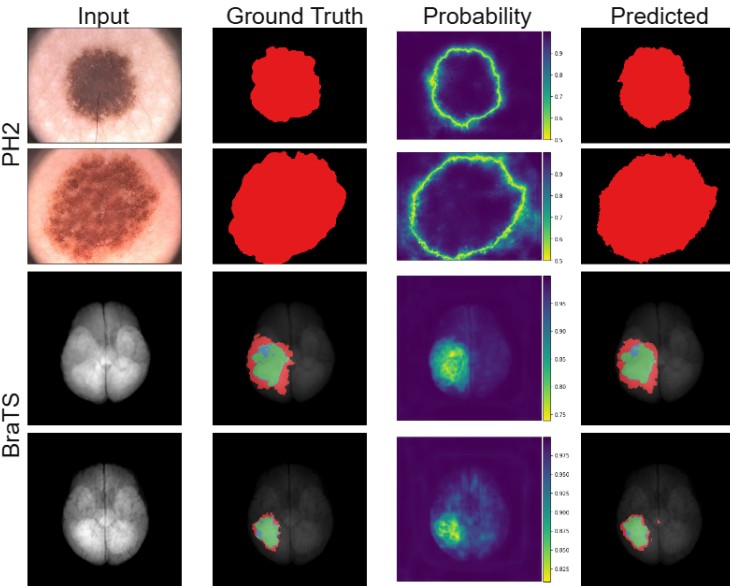

Figure 6: More segmented examples from the test datasets of PH2 and BraTS.

## C.4. BraTS

The data for the experiment is available at `http://medicaldecathlon.com/`. Only the training data is available for this task, consisting of 484 images. We split this dataset into training, validation, and test sets for our experiments.

The network begins with two convolutional stem blocks that use steerable convolutions to extract multi-resolution features. These blocks include non-linearities and batch normalization for stable training, as well as average pooling layers for downsampling. The features for each irrep are in a ratio of 8:4:2, with the number of features increasing in subsequent layers according to this ratio. The encoded features are then passed through a transformer encoder, which captures long-range dependencies and spatial relationships using multiple layers and positional encodings. The decoder consists of convolutional head blocks and trilinear interpolation kernel that upsample the features. The final output is generated by taking the absolute value of these complex features, corresponding to the constant representation.

The networks were trained using the Adam optimizer (Kingma & Ba, 2014), with an initial learning rate of $1 \times 10^{-2}$, which was reduced by a factor of 0.5 every 20 epochs, for a total of 100 epochs. A batch size of 1 was used, and training the largest model took 40 hours on a 16GB GPU. A larger batch size could not be used due to out-of-memory errors.

