# OpenReview forum: "Steerable Transformers for Volumetric Data"
_ICML.cc/2025/Conference — ICML 2025 poster_

### Official Review · Reviewer_URnE · 2025-02-27

**Overall Recommendation:** 3

**Summary:**

This paper introduces a new equivariant transformer architecture in two variants, with symmetry group either SE(2) or SE(3), utilizing a steerable (Fourier) basis. In numerical experiments, performance gains are shown if some layers in steerable CNNs are replaced by the novel attention layers.

## Update after rebuttals
I thank the authors for their thorough rebuttal and additional clarification on my question. That the scaling behavior of this setup is independent of the steerable nature of their architecture and therefore not a benefit of the proposed model is an important point which should be mentioned in the corresponding section. However, this is not a major contribution of this work.

I agree with the other reviewers that the tasks investigated in the experimental section are not very difficult, I see this paper more of a proof of concept of using steerable layers in transformers. This fills a hole in the literature, and does not need to be a new competitive architecture for some specific application domain.

For these reasons, I’m happy to maintain my score.

**Claims And Evidence:**

The paper claims that replacing steerable CNN layers by the novel steerable attention layers improves performance. This claim is tested on four diverse datasets. The performance benefits seem small but do lie outside of the standard deviation bands based on 5 runs.

**Essential References Not Discussed:**

None.

**Experimental Designs Or Analyses:**

The authors evaluate their architecture by replacing certain layers of steerable CNNs with their novel steerable attention layers. These combined architectures are not as common as pure CNN or pure attention architectures and I would find it more convincing to see a comparison of the latter.

**Methods And Evaluation Criteria:**

The datasets seem appropriate for evaluating the performance of this architecture.

**Other Comments Or Suggestions:**

I found these typos:
- No integral over SO(d) in (1)
- Are the indices correct on the RHS of $s_{ij}$ in section 3.2 (compare indices in $s_{ij}$ in section 2.1)?

**Other Strengths And Weaknesses:**

Strengths:
- Straightforward implementation of an equivariant transformer in a steerable basis
- Background is well-introduced
- Diverse datasets in numerical experiments

Weaknesses:
- No comparison to other equivariant transformers
- No comparison of pure steerable CNN vs pure steerable transformers

**Questions For Authors:**

Since transformers scale quadratically in the number of input tokens, I do not understand how the computational complexity of the steerable transformers can match the scaling of convolutions as claimed in section 3.5. Could you clarify?

**Relation To Broader Scientific Literature:**

Equivariant transformer architectures are an active area of research. Using (steerable) Fourier bases to construct equivariant architectures is not novel, but it seems that the most straightforward implementation of this, without tensor products and by using norm-nonlinearities as realized here, has not been done before. For this reason, I think this is a valuable baseline to consider.

**Theoretical Claims:**

The claimed equivariance of the layers proposed is straightforward to see in the steerable basis.

---

> ### Author Rebuttal · Authors · 2025-04-01
>
> We thank the reviewer for their detailed review and thoughtful suggestions. We have carefully considered the questions and concerns raised and provide detailed responses below.
>
> **Weaknesses**
>
> We have included comparisons with other methods on both the Rotated MNIST and ModelNet10 datasets in the supplementary material. Specifically, Table 3 of the Supplementary Material compares our approach against a range of equivariant methods on Rotated MNIST, including both convolutional and transformer-based architectures. For ModelNet10, while we were unable to find any existing equivariant transformer-based methods for direct comparison, Table 4 presents results alongside other equivariant approaches, covering both volumetric and point cloud-based techniques.
>
>
> In response to the reviewer’s critique regarding the lack of comparison between pure transformer-based and convolutional architectures, we conducted preliminary experiments on the Rotated MNIST dataset. The results are shown in Table 3 [at this anonymized GitHub link](https://anonymous.4open.science/r/Rebuttal-7B8B/table.pdf).
>
> In these models, we patchified the image and fed the resulting patches into a 3-layer transformer. The transformer output was then pooled and passed through fully connected layers for classification. The number of parameters for these standalone transformer models are approximately $1.10$M and $2.58$M for Fourier cutoffs of \(k = 4\) and \(k = 8\), respectively—comparable to the models in Table 1. Our findings show that reducing patch size improves accuracy but increases the runtime, and still falls short of the performance achieved by our hybrid model. This aligns with previous observations in literature (e.g., Xiao et al., 2021) that standalone ViTs require large datasets to outperform CNNs, and that convolutional encoders enhance transformer performance. These additional results offer a clearer comparison between standalone steerable transformers, steerable convolutional models, and our hybrid approach in the volumetric setting. We thank the reviewer for this insightful suggestion.
>
> Xiao et al. (2021). Early convolutions help transformers see better. Advances in Neural Information Processing Systems.
>
> **Questions**
>
> We agree with the reviewer that the complexity analysis warrants further clarification. The theoretical computational complexity of the transformer is $O(N^2C + NC^2)$, where $N$ is the number of patches and $C$ is the number of channels. The term $NC^2$ dominates in the regime $N \ll C$. As reviewer M9Pw correctly noted, this condition does not hold for raw image inputs—e.g., the BraTs brain MRI dataset has a resolution of $240 \times 240 \times 155$, so $N \ll C$ is clearly not satisfied at the pixel level.
>
> However, $N$ does not refer to the total number of pixels. In ViTs, $N$ represents the number of image patches after patchification: $N = \frac{ \text{Number of pixels}}{k^d}$, where $k$ is the patch size and $d$ is the image dimensionality. While $N$ can still be large for moderate $k$, our approach is not based on applying the transformer directly to raw image patches.
>
> Instead, we use a convolutional encoder to down-sample the input image before applying the transformer. This design is motivated by practical constraints: directly applying a transformer to high-resolution images requires a large patch size to stay within memory limits, which sacrifices local detail; conversely, using small patches leads to memory bottlenecks. Our hybrid convolution-transformer model addresses this by letting convolutions capture local features and the transformer focus on global context.
>
> For example, in our BraTs experiments, the image is down-sampled using two pooling layers (of sizes 8 and 4), resulting in a feature grid of $7 \times 7 \times 4$, i.e., $N = 196$ patches. The transformer operates on these features using $64$, $32$ and $16$ channels for the $\ell = 0, 1, 2$ components, respectively. Since each $\ell$-th component is a $(2\ell + 1)$-dimensional vector, this results in a total of 240 channels. In this setting, $N < C$, and our complexity calculations reflect this regime. Under these conditions, the computational cost of applying a transformer is comparable to that of a convolution.
>
> Finally, we also note that applying a sparse attention mask (restricting attention to nearby patches, as in graph-based models) is indeed a viable way to reduce complexity. However, such an approach undermines the main strength of transformers—capturing long-range dependencies. If only local interactions are needed, convolutions are a more natural and efficient choice.

---

> > ### Comment · Reviewer_URnE · 2025-04-03
> >
> > Thank you for the additional results and clarifications about the scaling behavior.
> >
> > Do I understand correctly that the favorable scaling behavior you describe is due to the patching combined with the convolutional encoder? It seems that this is a general feature of combined convolutional- and transformer architectures and not something which is specific to the steerable transformer introduced in this paper. If I’m mistaken about this and there are scaling benefits from the steerable architecture, I’d appreciate a short reply.

---

> > > ### Author Response · Authors · 2025-04-04
> > >
> > > We thank the reviewer for the thoughtful follow-up and for raising this important point.
> > >
> > > You are correct that the favorable scaling behavior we describe primarily arises from the use of patching in combination with a convolutional encoder. This is a general feature of transformer-based architectures and is not specific to the equivariant aspects of our model. This strategy is common to many hybrid convolution-transformer architectures as it effectively reduces the input resolution before feeding it into the transformer without relying on excessively large patches, thereby easing the computational burden.
> > >
> > > The steerable transformer introduced in our work does not offer additional scaling advantages beyond this architectural strategy. Instead, our goal was to bring the benefits of steerable  representations into the transformer framework, enabling equivariance while maintaining practical scalability through architectural design.
> > >
> > > We sincerely thank the reviewer for their detailed feedback and thoughtful suggestions. We appreciate the time and effort you dedicated to reviewing our submission.

---

### Official Review · Reviewer_GYnN · 2025-03-11

**Overall Recommendation:** 3

**Summary:**

The authors present an SE(3) steerable attention layer. Specifically, input features are assumed
to be equivariant tensor features in the specific form as is output by SE(3) steerable networks
of (Weiler et al. 2018-2019). The authors key contribution could be understood as a novel form of
positional embedding which when passed through the standard attention layer (modified only to compute
the dot product between complex valued features), preserves SE(3) equivariance. Comparisons are performedonly against the relatively dated steerable convolutions proposed by (Weiler et al. 2018-2019)

**Claims And Evidence:**

The authors claim to present an attention layer which preserves the specific type of SE(3) equivariance
enjoyed by the features of the type in (Weiler et al. 2018-2019). The authors provide a proof which to my
inspection is sound and does verify their claim.

The authors also claim that their steerable attention layer improves the performance of existing
SE(3) steerable convolution networks. While the reported results make this claim technically correct,
the performance increase is at best marginal and several of the datasets considered are toy. More generally,
in the supplement, the results of additional methods not based on the method of (Weiler et al. 2018-2019) are shown and the authors proposed approach performs significantly worse.

**Essential References Not Discussed:**

The authors do not discuss Vector Neurons (Deng et al. 2021) which have been show to far outperform
other equivariant nonlinearities, and in particular those discussed and employed by the authors.
It appears that with a small modification, this nonlinearity could be used with the
authors approach (since \rho(R) are orthogonal).

**Experimental Designs Or Analyses:**

See above.

In general, the results are not compelling and do not convince me that the proposed method offers a
meaningful improvement over existing methods.

**Methods And Evaluation Criteria:**

The authors proposed method appears to be technically sound, if limited. The evaluation is
extensive enough, though the results are not compelling.

One of the chief limitations it that it appears the method cannot "patchify" volumetric inputs
as is standard in vision transformers -- i.e. that it is assumed to operate with a patch size of 1^3
and therefore likely scales poorly. Patchification is key to the success of vision transformers, and the
inability of the proposed method to do this is a significant weakness.

**Other Comments Or Suggestions:**

See above.

**Other Strengths And Weaknesses:**

This paper is well written, but has has several important weaknesses:

The novelty of the method is limited -- the authors real contribution appears to be the positional embedding,
it is already well known that orthogonally-equivariant features make the standard attention mechanism equivariant.

Moreover, while the title of the paper implies that the authors are presenting a novel transformer, they are really just
presenting a specialized positional embedding. Important practical considerations, like an equivariant patchification method, are not considered and limit the scalability of the method.

In general, the results are not compelling. The steerable convolutions proposed by (Weiler et al. 2018-19) are now
far from state of the art (as shown in the results reported by the authors in the supplement). This method offers what appears to bea way to marginally improve their performance, but does not come close to state-of-the-art.

In addition see the comments on nonlinearities above.

**Questions For Authors:**

N/A

**Relation To Broader Scientific Literature:**

The key contributions are unclear. What the authors present is a specialized attention mechanism
designed to work on a specific set of features output by the now dated methods of (Weiler et al. 2018-19)
which have scalability issues and are now far from state of the art. Furthermore, the authors proposed approach
does not improve the performance steerable convolutions to be comparable to existing state of the art methods.

**Theoretical Claims:**

The authors main theoretical claim is that their proposed attention layer is equivariant, the the
provided proof appears to be correct.

---

> ### Author Rebuttal · Authors · 2025-04-01
>
> We thank the reviewer for their detailed review and thoughtful suggestions. We have carefully considered the questions and concerns raised, and provide detailed responses below.
>
> **Patchification**
>
> We would like to clarify the reviewer’s concern regarding "patchification". In ViTs, patchification refers to dividing an image into patches, flattening them, and projecting them via a learnable linear embedding. This operation is equivalent to using a convolutional layer with stride equal to kernel size. In fact, most ViT implementations, implement patchification using a strided convolution. This perspective aligns with the original ViT paper (Dosovitskiy et al., 2020), which states:
>
> >"As an alternative to raw image patches, the input sequence can be formed from feature maps of a CNN...As a special case, the patches can have spatial size 1x1, which means that the input sequence is obtained by simply flattening the spatial dimensions of the feature map and projecting to the Transformer dimension."
>
> So, patchification can be thought of as a special case of a convolutional encoder. These downsample the volumetric input to a grid of lower resolution tokens which are then passed into the transformer. However, we did not emphasize this in the manuscript as it is not a novel take on ViTs. Thus, contrary to the reviewer’s concern, our method does support patchification in a manner consistent with standard ViTs, adapted appropriately to the steerable and volumetric setting.
>
> **Comparison with Weiler et al**
>
> We appreciate the reviewer’s feedback and the opportunity to clarify our contributions. Our use of steerable features is not a design choice but is motivated by theoretical foundations. As shown by Cohen et al. (2017) and Kondor et al. (2018), steerable convolutions (Weiler et al., 2018–19) represent the most general class of equivariant linear maps. Our work builds on this foundation by introducing a transformer architecture that operates on these structured feature fields. While steerable convolutions may face scalability limitations, they remain theoretically sound and widely adopted. Our aim is to enhance, not replace, their capabilities with global attention.
>
> We would like to clarify the reviewer’s comment regarding the performance of steerable convolutions.Table 3 of our supplementary material compares our method with other equivariant approaches on Rotated MNIST. To the best of our knowledge, Weiler et al. (2019) still represents the state of the art on this benchmark. It is unclear which specific work the reviewer is referring to in stating that Weiler et al. (2018–19) are far from state of the art. If the reviewer is aware of more recent methods with better performance, we would be grateful for references and will gladly include them in the revision.
>
> **Vector Neurons**
>
> We thank the reviewer for pointing out Vector Neurons (Deng et al., 2021), but we must emphasize that the methodology proposed in that work operates in a fundamentally different setting from ours. Vector Neurons are specifically designed for point cloud data, where inputs are of the form $V\in\mathbb{R}^{N\times 3}$, ($N$=number of points). In contrast, our work focuses on volumetric data of the form $\mathbb{R}^{C\times K\times K\times K}$, ($K$=resolution and $C$=channels).
>
> It is not clear to us how the Vector Neurons framework can be applied to the volumetric setting. For instance, their linear layer (Sec. 3.1) involves right-multiplication of the input by a learnable matrix. A comparable operation in our setting would correspond to a convolution with filter os size $1^3$-known to have limited expressiveness. Furthermore, their nonlinearity (Sec 3.2) operates directly on the outputs of such a linear transformation, and is defined as:
>
> $v'=q\text{ if }q^Tk\geq0$
>
> $=q-(q^Tk)k/||k||^2\text{ else where}\quad q=WV,k=UV$
>
> This relies on the comparison of the inner product value with zero, which is not meaningful in our setting where features lie in complex space due to the Fourier-based representation. Consequently, applying this nonlinearity in our framework would not be straightforward,. We are unsure by what the reviewers mean by small modifications in this context. It is also unclear to us why this nonlinearity is expected to outperform norm-based ones (Worrall et al., 2017), which have been well validated for 2D data. In contrast, to the best of our knowledge, the Vector Neurons nonlinearity has only been evaluated on point cloud datasets (ModelNet40, ShapeNet) and has not been demonstrated in datasets we evaluate. This makes direct comparison difficult and potentially misleading. Given these differences, we do not see a viable path for incorporating the Vector Neurons nonlinearity without substantially redefining the core of our framework. We would be appreciate if the reviewer could clarify how they envision Vector Neurons being adapted to our setting, and what specific advantages are expected over established volumetric methods.

---

> > ### Comment · Reviewer_GYnN · 2025-04-03
> >
> > Thanks for a refreshing rebuttal -- I appreciate and welcome the authors vigorous pushback on my concerns.
> >
> > The authors make a good point regarding patchification that I did not previously realize and which addresses my concern. That said, it would be nice to mention the interpretation of equivariant convolution as patchification in a sentence or two in the manuscript to help readers understand the analogy with VITs.
> >
> > Regarding vector neurons, it appears that the authors proposed features at each token $f$ are complex $d_\rho \times d_m$ features as in Eq (5), and transform via a unitary representation $f(R x_i + t, \rho) \mapsto \rho(R) f(x_i, \rho)$. Assuming this understanding is correct (if not, please clarify as the only explicit mention of how features are expected to transform under SE(3) is on line 190 and I think this should be more explicit, considering you're proposing an equivariant network), then a simple generalization of vector neurons that would act pointwise via learned matrices could be implemented as follows
> > $$ Q = f W_Q, \quad K = f W_K $$
> > $$
> > f' = Q - \frac{\gamma(\textrm{tr}(Q^\dagger K))}{\textrm{tr}(K^\dagger K)} K
> > $$
> > where $\gamma$ is any choice of activation function for complex numbers. For example, $\gamma(x)$ could be defined to return $0$ if $\Re(x) < 0 $ and $x$ otherwise. Vector neurons were proposed precisely to address the weaknesses of (now dated) norm-based nonlinearities when dealing with features that transform under orthogonal/unitary transformations. While the original paper focuses on 3D vector-valued data, the concepts are easy to generalize to other types of features and have gained wide adoption due to their efficacy. I encourage the authors to experiment with them in their work.
> >
> > My two major concerns are still outstanding and are discussed as follows:
> >
> > 1). Weakness of experimental results.
> > The experiments as they stand now do not show that the proposed approach materially improves upon steerable convolutions. Benchmarking results on rotated MNIST and rotated ShapeNet are meaningless in 2025  as those datasets are trivial and the field has moved on to more challenging tasks. While I am not well versed in the medical image literature, it looks like reviewer M9Pw also points out that PH2 is also essentially a meaningless dataset for benchmarking purposes. Additionally, they echo my concern in that there are no comparisons to domain specific methods.
> >
> > If the authors are serious about demonstrating the applicability and usefulness of their approach, then I would recommend applying their method to a more challenging task. For instance, voxelized representations are at the core of SoTA 3D generative models (https://meshformer3d.github.io/). If the authors can show that their method can be used a drop in replacement for the convolutions in such an architecture, and either improves outcomes and/or provides new functionality, then I would consider that experiment alone to be compelling evidence.
> >
> > 2). Novelty of the the contribution.
> >
> > As far as I can tell, aside from conceptualization, the only truly original technical contribution of the authors is the proposed positional encoding. Otherwise, the transformer is just a standard transformer modified to work with complex features, and requires steerable convolutions for pacification. This positional embedding could perhaps be a valuable contribution, but that isn't the way the paper is framed, nor do the experiments actually validate its effectiveness.
> >
> >
> > As a more general comment, I think by now the bar has been raised for equivariant learning and showing marginally more effective results on trivial datasets is no longer enough to clear it. If there only exist trivial datasets and benchmarks for the authors chosen problem, then the onus is on them to find a more compelling or previously overlooked application which highlights the effectiveness of their proposed approach.
> >
> > I welcome and look forward to responding to any additional comments by the authors.
> >
> >
> > **Edit**: After some reflection I feel I have judged this paper too harshly.  At this point, I think my previous concerns about the papers novelty are unfounded, and that the paper provides a nice technical contribution. I think the evaluations are still an outstanding issue, but otherwise I am OK with the paper. I will raise my score accordingly.

---

> > > ### Author Response · Authors · 2025-04-04
> > >
> > > We thank the reviewer for the thoughtful follow-up and for highlighting this important point.
> > >
> > > We appreciate the suggestion to investigate Vector Neuron nonlinearities and plan to include them in future experiments as part of our ongoing work on steerable architectures.
> > >
> > > We acknowledge that the current experiments do not fully capture the scale and complexity of large real-world computer vision tasks. Due to limited computational resources, we were unable to extend our experiments beyond the datasets presented. Our goal in this work was to make a technical contribution to the theory of steerability, with the experimental section serving primarily as a proof of concept.
> > >
> > > We also appreciate the suggestion to pursue more complex tasks, such as 2D-to-3D image generation, as demonstrated in Meshformer. However, we note that Meshformer was trained using 8 H100 GPUs, which exceeds the computational resources available to us. We validated our model within the practical constraints at our disposal. That said, we agree that the full value of our approach will be best realized in large-scale biomedical imaging contexts, and we see scaling up this architecture for applications such as medical image segmentation as a promising direction for future work.
> > >
> > > We sincerely thank the reviewer for their detailed feedback and thoughtful suggestions. We appreciate the time and effort you dedicated to reviewing our submission.

---

### Official Review · Reviewer_M9Pw · 2025-03-13

**Overall Recommendation:** 2

**Summary:**

This paper explores the steerable (SE3 equivariant) vision transformer for volumetric data. The proposed framework was built upon existing steerable convolution and vision transformers. Steerable transformers was evaluated on 2D and 3D classification (rotated MNIST and ModelNet10) and segmentation (skin images and brain MRI) and outperformed the original steerable convolution. In addition to that, additional comparisons with previous studies spanning from 2015 to 2019 were conducted on ModelNet10 with z rotation perturbation.

The contribution was claimed to integrate ViT and steerable convolutions to yield steerable transformers for 3D volumetric image data (instead of a point cloud, which was extensively studied).

**Claims And Evidence:**

1. The claim of equivariance was supported by Proposition 3.1.
2. The claim regarding complexity analysis (Section 3.5) was not supported by convincing evidence; please refer to the Theoretical Claims section.
3. Introducing transformers' attention mechanism enhances the baseline performance of steerable convolution.

**Essential References Not Discussed:**

Some highly relevant and domain-specific previous studies are not discussed.

[1]: Sangalli, Mateus, et al. "Moving frame net: SE (3)-equivariant network for volumes." NeurIPS Workshop on Symmetry and Geometry in Neural Representations. PMLR, 2023.
[2]: Kuipers, Thijs P., and Erik J. Bekkers. "Regular se (3) group convolutions for volumetric medical image analysis." International Conference on Medical Image Computing and Computer-Assisted Intervention. Cham: Springer Nature Switzerland, 2023.

**Experimental Designs Or Analyses:**

The experimental designs and analyses are not comprehensive.

I am unsure why the authors included two 2D and two 3D datasets in the experiments. The main aim of this paper is to integrate steerable conv with ViT for volumetric data. Therefore, there should be a focus on experiments on 3D volumetric datasets (which were dominated by medical images in the real world). Additionally, a few studies focused on volumetric medical images should be compared, including [1] and [2].

I also want to point out that, when comparing to other Equivariant Attention methods in Table 3, α-R4 CNN (Romero et al., 2020) has only around 75k params, and GE-ViT (Xu et al., 2023) has only 45k params, compared to around 1M params of the proposed steerable transformer. While the authors argued in lines 901-902 that the performance difference among E2CNN (Weiler & Cesa, 2019), Weiler et al. (Weiler et al., 2018b), and the proposed steerable transformers come from the computation limitation, it is not explained in the current manuscript that the proposed framework is also considerable larger than GE-ViT and α-R4 CNN.

[1]: Sangalli, Mateus, et al. "Moving frame net: SE (3)-equivariant network for volumes." NeurIPS Workshop on Symmetry and Geometry in Neural Representations. PMLR, 2023.

[2]: Kuipers, Thijs P., and Erik J. Bekkers. "Regular se (3) group convolutions for volumetric medical image analysis." International Conference on Medical Image Computing and Computer-Assisted Intervention. Cham: Springer Nature Switzerland, 2023.

**Methods And Evaluation Criteria:**

While the proposed methods make sense, the main discovery of this paper seems to be how to maintain the equivariance of the output from a steerable convolution throughout a vision transformer via some slight modifications to position encoding and nonlinear activation functions.

The evaluation criteria of classification and segmentation performance are standard.

**Other Comments Or Suggestions:**

N/A

**Other Strengths And Weaknesses:**

Other strengths:
1. The presentation was good
2. The discussion of existing works was extensive and comprehensive while lacking acknowledgment of a few domain-specific papers.

**Questions For Authors:**

1. The visualization of the BraTs dataset looks weird and does not look like a brain MRI; how was it generated?
2. How do we interpret the probability map for BraTs datasets shown in Fig. 3 and 5? It seems to have no correlation with predicted labels.
3. Why is the parameter number of other comparing methods not listed in Table 3? The α-R4 CNN (Romero et al., 2020) has only 73.13k params compared to ~ 1M params of the proposed Steerable Transformer. And GE-ViT (Xu et al., 2023) has only around 45k parameters.
Is the performance gain coming from the difference in parameter number in this case (i.e., proposed vs. GE-ViT and α-R4 CNN).

**Relation To Broader Scientific Literature:**

Previous studies mostly focused on equivariant networks on 2D and 3D point cloud analysis. This paper focuses on designing SE(3)-equivariant ViT for volumetric image data, which has practical value in volumetric medical image analysis.

**Theoretical Claims:**

I checked the correctness of theoretical claims. I found the complexity analysis (Sec 3.5, lines 310-329) has flaws. Specifically, when analyzing the complexity of the proposed steerable attention, the author ignores a significant item using an assumption that almost does not exist in the real world. The original manuscript writes, and I quote (lines 314-316), "SA has a complexity $\mathcal{O}(NC^2+N^2C)$, and FFN adds $\mathcal{O}(NC^2)$, totaling $\mathcal{O}(NC^2+N^2C)$. **When $N<<C$, the $\mathcal{O}(NC^2)$ term dominates.**" The authors then conclude that (Lines 323-326), "In summary, with a fixed kernel size, the computational complexity of steerable attention matches that of convolution."

This **When $N<<C$ assumption**, unfortunately, does not exist in many real-world 3D volumetric data, especially in medical images. The BraTs brain MRI dataset used in this study has a matrix size of 240x240x155, and real-world high-resolution chest CT may have a size of 512x512x300+; in either case, the final sequence length fed into ViT after the steerable convolution would not have an $N<<C$.

The correct time complexity of steerable transformer, waiving that unrealistic assumption, would be $\mathcal{O}(Nd_{\rho}^2C^2+N^2d_{\rho}C)$, not as claimed in Line 318 $\mathcal{O}(Nd_{\rho}^2C^2)$. Compared to $\mathcal{O}(Nd_{\rho}^2C^2k^d)$ of steerable convolution (note that N for ViT is sequence length and N for conv is pixel number), it is clear that steerable transformer has higher complexity for real-world volumetric data.

Therefore, the time complexity analysis has flaws, and to better evaluate the complexity, it is better to use other practical metrics, such as training and inference throughput time per fixed batch, floating-point operations per second, or multiply-accumulate operations.

---

> ### Author Rebuttal · Authors · 2025-04-01
>
> We thank the reviewer for their detailed review and thoughtful suggestions. We have carefully considered the questions and concerns raised, and provide detailed responses below.
>
> **Complexity Calculations**
>
> We acknowledge the reviewer’s concern regarding the complexity calculation. It is true that the assumption $N \ll C$ does not generally apply to raw image inputs. However, in our case, $N$ denotes the number of patches in the *downsampled* feature map produced by the convolutional encoder—not the number of pixels in the original image. For example, in our BraTs experiments, two pooling layers (with sizes 8 and 4) reduce the input to a feature map of dimensions $7 \times 7 \times 4$, yielding $N = 196$ patches. The transformer then operates on this representation using 64, 32, and 16 channels for the $\ell = 0, 1, 2$ components, respectively, totaling 240 channels (each $\ell$ component is a $2\ell + 1$-dimensional vector). In this configuration, $N < C$, and our complexity analysis reflects this setting, where the computational costs of attention and convolution are comparable. Please refer to our response to reviewer URnE for a more detailed discussion.
>
> In line with the reviewer’s suggestion, we have also updated Table 1 to report both training and inference times for each model. The revised table is available in Table 1 [at this anonymized GitHub link](https://anonymous.4open.science/r/Rebuttal-7B8B/table.pdf). The results show that steerable transformers and steerable convolutions with similar parameter counts have comparable runtimes.
>
> **Experimental Design**
>
> The framework presented in our manuscript is designed to operate in general $d$ dimensions, encompassing both 2D and 3D settings. While our discussion of group representations are motivated by the 3D setting, we believe that the proposed approach is also novel in the 2D case. To our knowledge, prior work on equivariant vision transformers for image data has not employed this formulation. For this reason, we have demonstrated our method in both 2D and 3D contexts.
>
> We agree with the reviewer that medical imaging applications is particularly well-suited for this architecture. In response to this comment, we will expand the manuscript to include more such experiments and discussions regarding relevant works, including those by Sangalli et al. (2023) and Kuipers et al. (2023). We appreciate the reviewer for pointing us in this direction.
>
> **Question 1 and 2**
>
> We thank the reviewer for the feedback on our visualization. The BraTs dataset has resolution $4\times 240\times 240\times 155$, where the first dimension corresponds to four MRI modalities (T1, post-contrast T1-weighted, T2, and T2-FLAIR) rather than traditional channels. In our initial submission, we averaged over the modality and the first spatial dimension, followed by a 90-degree rotation:
> ```python
> inputs_plot = torch.rot90(torch.mean(inputs, dim=(0,1)))
> probs_plot = torch.rot90(torch.max(probs, dim=dim)[0])
> ```
> However, we acknowledge that this view offered limited insight. To improve interpretability, we revised the visualization as follows:
> - Display only the T1 modality.
> - Use the transverse (z-axis) view by averaging over the third spatial dimension.
> - For probabilities, take the mean across the last spatial dimension.
>
> Updated code:
> ```python
> inputs_plot = torch.rot90(torch.mean(inputs[0], dim=2))
> probs_plot = torch.rot90(torch.mean(probs, dim=2))
> ```
> The improved visualizations are available [at this anonymized GitHub link](https://anonymous.4open.science/r/Rebuttal-7B8B/brats.png). We appreciate the reviewer’s suggestion, which helped enhance the clarity of our presentation.
>
>
> **Question 3**
>
> While some equivariant Vision Transformers (ViTs) report lower parameter counts, they are typically much more memory-intensive. For instance, Romero et al. (2020) noted that
> > “the $\alpha$-p4 All-CNN requires approximately 72GB of CUDA memory, as opposed to 5GB for the p4-All-CNN. This is due to the storage of the intermediary convolution responses required for the calculation of the attention weights.”
>
> Romero et al. (2021) also reported slow training times, even with multiple GPUs. Since GE-ViT (Xu et al., 2023) builds on this framework, it expected to face similar challenges. We believe the reduced parameter count in such models is a consequence of significant memory and compute overhead. In contrast, our method strikes a balance by augmenting steerable convolutions with a transformer applied only to down-sampled features, greatly reducing computational demands. GGiven this, we compare our model with both equivariant attention-based and convolutional methods. For example, Weiler et al. (2021) uses ~6M parameters, which we consider a fair baseline. We acknowledge that omitting the parameter count in the original manuscript was an oversight. The revised table is available in Table 2 [at this anonymized GitHub link](https://anonymous.4open.science/r/Rebuttal-7B8B/table.pdf).

---

> > ### Comment · Reviewer_M9Pw · 2025-04-03
> >
> > I appreciate the effort and time the authors invested in preparing for the rebuttal. Thank you!
> >
> > First of all, I'd like to apologize for a few missing words from my initial review in Section Questions For Authors No.3. Fortunately, it did not alter the meaning of that point, and I modified my review to correct it.
> >
> > Unfortunately, I still have concerns regarding the claims about time complexity and Experimental Design.
> >
> > **Time complexity**:
> >
> > 1. N is the number of patches. I explicitly stated this in my initial review:
> > > note that N for ViT is sequence length and N for conv is pixel number
> >
> > Clearly, sequence length is the number of patches. My point is, given the large volumetric matrix, there is no $N<<C$ in the medical image domain. The authors replied:
> > > It is true that the assumption $N<<C$ does not generally apply to raw image inputs.
> >
> > In fact, just in the same paragraph, with N=196, C=240, and authors stated that $N<C$, I'd like to reiterate my point: the assumption $N<<C$ does not apply to volumetric medical images with size larger than 240x240x155, $N<<C$ is just not applicable to medical images in the raw image space **and** in the patches/sequences space.
> >
> > In my initial review, I pointed out the example of CT scans. Some chest CTs have a size of 512x512x300+, which is about 2560 patches, and might be considered as $N>>C$.
> >
> > **Benchmarking**:
> >
> > This paper was submitted as Application-Driven Machine Learning. With medical images as the primary volumetric data in the real world and having significant practical value, the current experiments are insufficient to support this manuscript's application value. I agree with reviewer GYnN that some datasets are toy datasets. One of the medical image datasets, PH2, only contains **200** 2D images released in 2013 [1], which is really too outdated and too limited. It is hard to imagine using a dataset with 200 medical images in 2025...
> >
> > As the datasets are not strong enough, and comparisons with domain-specific methods are not thorough, I think the current results, together with the rebuttal, can't support and justify the application value of the proposed work. Therefore, I decided to maintain my initial rating of weak reject.
> >
> >
> > [1]: T. Mendonça, P. M. Ferreira, J. S. Marques, A. R. S. Marcal and J. Rozeira, "PH2 - A dermoscopic image database for research and benchmarking," 2013 35th Annual International Conference of the IEEE Engineering in Medicine and Biology Society (EMBC), Osaka, Japan, 2013, pp. 5437-5440, doi: 10.1109/EMBC.2013.6610779.
> >
> > ---**Apr. 6 update**---
> >
> > Thank the authors for their keen effort in rebuttal, and I appreciate the meaningful discussion. Unfortunately, the latest reply on Apr. 4th did not resolve my concern about benchmarking, and additionally, I found some false statements in it. Therefore, I stand with my initial rating: 2: Weak reject (i.e., leaning towards reject, but could also be accepted, *if everyone is okay with this weak evaluation*).
> >
> > A quick **facts check** for some **FALSE** information I found in the rebuttal reply:
> > 1. MedMNIST (initial version, [preprint](https://arxiv.org/abs/2110.14795v1) 27 Oct 2021, low resolution of 28x28), Table 2: there are 12 subsets totaling 708,069 images, on average **59k** image per subset. **NOT** 1,500 images per subset as claimed by the authors. The dermatology image subset in MedMNIST contains 10,015 images, vs. 200 in PH2.
> > 2. MedMNIST+ (high resolution released on Jan. 2024, reference link to [github](https://github.com/MedMNIST/MedMNIST/commit/f49eb7d4c4eb95b623f47764504868c30c23f0b4#diff-47a81cfe1782b6bbcdd0c521c553cc1d133dbb63789dc1156e88400f442efec3)). There is an updated MedMNIST with larger sizes: 64x64, 128x128, and 224x224 for 2D, and 64x64x64 for 3D. So MedMNIST is not low-resolution only.
> >
> > If the authors still stand with their statement about MedMNIST, I would appreciate a **reference** for double-checking.
> >
> > Even if the authors do not want to include MedMNIST, [ISIC](https://www.isic-archive.com/) is a much more convincing dataset than PH2.
> >
> > A quick reply to complexity: more aggressive downsampling/pooling comes with degraded performance, meaning when the framework is applied to more real-world high-resolution data, it won't perform as well as presented here.
> >
> > I want to reiterate some weaknesses of the experiment and evaluation that are fundamental and seem unfixable in the scope of this rebuttal:
> >
> > 1. Not convincing dataset: PH2, containing only 200 images.
> > 2. Not compared to previous domain-specific works (this is listed as a weakness because this paper was submitted as Application-driven machine learning).
> >
> > ---**End of Apr. 6 update**---

---

> > > ### Author Response · Authors · 2025-04-04
> > >
> > > We thank the reviewer for the thoughtful follow-up and for raising this important point. We would like to further clarify some of the points raised by the reviewer.
> > >
> > > **Complexity:** We would like to clarify that for the complexity calculation to hold, it suffices to have $N \asymp C$, rather than requiring $N \ll C$. The key point we aimed to convey in our earlier response is that $N$—the number of tokens—is determined by the architectural design (i.e., how we downsample the input). In practice, for large medical images such as CT scans, it is standard to apply aggressive downsampling (e.g., through pooling or strided convolutions) to reduce computational burden before feeding the data into a transformer.
> > >
> > > All of our experiments were conducted under this assumption: that the input is compressed sufficiently so that $N$ and $C$ are of comparable magnitude. If the input resolution increases—for example, with a $512 \times 512 \times 300$ volume—we simply adapt the downsampling accordingly. Using two pooling layers of size 8 (instead of 8 and 4 as in we used in the BraTS experiment) would yield a feature map of size $8 \times 8 \times 5$, giving $N = 320$, which remains in the regime $N \asymp C$ with the same transformer configuration. In summary, whether we are in the $N \gg C$ or $N \asymp C$ regime is not determined by the input image size but by the architectural choices we make to process it. Our complexity analysis assumes $N \asymp C$, consistent with the design of our models.
> > >
> > > **Benchmarking:**
> > > We agree that the experiments we conducted are relatively small in dataset size—for example, 200 images in the PH2 dataset and 400 in the BraTS dataset. However, these datasets are challenging due to the high resolution of each image: PH2 images are $578 \times 770$, and BraTS volumes are $240 \times 240 \times 155$. These datasets actually reflect real life complexity of problems, where resolution is high and number of labelled data is low. In contrast, the domain-specific papers the reviewer referred to used the MedMNIST dataset, which contains around $1,500$ images per subset, but at significantly lower resolutions—$28 \times 28$ for 2D and $28 \times 28 \times 28$ for 3D. From this perspective, we believe our experiments better reflect the size and complexity of real-world biomedical imaging tasks. Due to limited computational resources, we were unable to extend our experiments beyond the datasets presented. Our goal in this work was to make a technical contribution to the theory of steerability, with the experimental section serving primarily as a proof of concept. We would also like to note that while the PH2 dataset was introduced in 2013, it continues to be used regularly in recent methodological work and remains a standard benchmark in biomedical image analysis [1,2].
> > >
> > > We sincerely thank the reviewer for their detailed feedback and thoughtful suggestions. We appreciate the time and effort you dedicated to reviewing our submission.
> > >
> > > [1] Rashid, Akram Arslan, et al. ''Segmentation and Classification of Skin Lesions Using Hybrid Deep Learning Method in the Internet of Medical Things."
> > >
> > > [2] Mirikharaji, Zahra, et al. "A Survey on Deep Learning for Skin Lesion Segmentation." Medical Image Analysis.

---

### Official Review · Reviewer_skHN · 2025-03-15

**Overall Recommendation:** 3

**Summary:**

This work introduces a steerable equivariant transformer operating in the spectral domain. The method calculates attention scores exclusively between Fourier embeddings associated with matching irreducible representations (irreps). These scores are then combined linearly to construct the final attention matrix, which is used to generate output tokens. The architecture is mathematically proven to maintain equivariance by design and demonstrates superior performance over baseline methods in both 2D and 3D tasks.

**Claims And Evidence:**

**Claim 1: Design of Steerable Transformer:**  Authors provided proof of the equivariance of the model. However, this claim will be bolstered if authors can empirically report the equivariance/invariance error (see Sec 6 [1]).

**Claim 2: Performance gain:** Works shows performance gain against steerable convolution. However, authors should also report the consistency of the model over the orbit of an input for each dataset. For example, in the case of ModelNet10, any rotation should not change the predicted probability scores.


[1] Scale-Equivariant Steerable Networks

**Essential References Not Discussed:**

N/A

**Experimental Designs Or Analyses:**

The experiments are conducted on four different datasets. As I mentioned earlier, additional metrics, such as equivariance error and consistency over the orbit of an input, should be reported for completeness.

**Methods And Evaluation Criteria:**

The chosen datasets are valid. However, as mentioned earlier, additional evaluation metrics, such as equivariance error

**Other Comments Or Suggestions:**

N/A

**Other Strengths And Weaknesses:**

The positional encoding design would benefit from greater clarity and intuition. While the authors provide a mathematical proof validating their approach, this formal validation should be preceded by a more accessible conceptual explanation.

I strongly recommend including a visual representation—specifically, a figure illustrating the positional encoding mechanism for a simple 2D case.

**Questions For Authors:**

N/A

**Relation To Broader Scientific Literature:**

The key contribution of the paper is the proposal of an equivariant transformer for volumetric data. Even though there exist models of equivariant transformers for vision, i.e., 2D data, the work proposes a more general framework for such transformers, which I believe will be of great interest to the machine learning community.

**Theoretical Claims:**

Yes. I went through the proof, and I found it to be correct.

---

> ### Author Rebuttal · Authors · 2025-04-01
>
> We thank the reviewer for their detailed review and thoughtful suggestions. We have carefully considered the questions and concerns raised, and provide detailed responses below.
>
> **Equivariance Error**
>
> We thank the reviewer for the insightful suggestion about equivariance error. In response, we have included a visual analysis of the equivariance error, following a methodology similar to that of Sosnovik et al. (2020). Specifically, we evaluated our model on randomly selected samples from the MNIST and ModelNet10 datasets, along with their rotated counterparts. The equivariance error is measured using the following expression:
>
> $$\Delta =\frac{||M(T_\theta[f]) - T_\theta[M(f)]||^2}{||M(T_\theta[f])||    ||T_\theta[M(f)]||},$$
>
> where $M$ denotes the model and $T_\theta$ represents a rotation by $\theta^\circ$. For each input image $f$, we computed $\Delta$ across $\theta \in [0^\circ, 90^\circ]$ in $5^\circ$ increments, averaging the results over 50 random samples per dataset. For the ModelNet10 dataset, we considered two rotation types: around the $Y$-axis and around the $Z$-axis.
>
> The resulting plots of $\Delta$ as a function of $\theta$ for varying numbers of transformer layers are available [at this anonymized GitHub link](https://anonymous.4open.science/r/Rebuttal-7B8B/Equivariance_error.ipynb). We observe that the equivariance error grows slightly with the number of layers, eventually stabilizing at approximately 1.5\% for Rotated MNIST and 10\% for ModelNet10.
>
> Sosnovik at al., (2020). Scale-Equivariant Steerable Networks. The International Conference on Learning Representations.
>
> **Positional Encoding Intuition**
>
> We would like to clarify the intuition behind the positional encoding formulation. The central idea is to encode not only the relative distance between two patches but also their relative direction. Since we use relative positional encoding, the encoding between two patches $x_i$ and $x_j$ is based on the difference vector, i.e., $P(x_i - x_j)$. To incorporate directional information, in a steerable setting, we extend this to include an angular component: $P(x_i - x_j, \theta)$ ($\theta\in (0,2\pi)$ in 2D case). This formulation is inherently translation invariant by construction. Additionally, we require the encoding to behave consistently under rotations. That is, if the input image is rotated by an angle $\phi$, $x \mapsto R_\phi x$, the relative position vector undergoes the same rotation, and in this case, the angular component of $P$ should shift accordingly, resulting in the transformation $P(x, \theta) \mapsto P(R_\phi x, \theta + \phi)$. In steerable neural networks, we operate in the Fourier domain, where such angular shifts correspond to modulations by Fourier basis functions. Specifically, the positional encoding $P(x, k)$ transforms under rotation as $P(x, k) \mapsto e^{-\iota k\phi} P(R_\phi x, k)$, reflecting the equivariance properties of the Fourier basis. This insight motivates our design choice in Equation (6), where we define the encoding as $P(x, k) = \phi(r, k) e^{-\iota k \theta}$, with $r = ||x||$ and $\theta$ denoting the polar angle of $x$. The function $\phi(r, k) \propto r^{-2}$ modulates the influence of distance, assigning greater weight to nearby patches. This formulation extends naturally to higher dimensions, leading to the 3D generalization presented in Equation (7).
>
> We appreciate the reviewer’s suggestion and have created a visualization to better convey the intuition behind our equivariant positional encoding (available [at this anonymized GitHub link](https://anonymous.4open.science/r/Rebuttal-7B8B/positional.png)).

---

### Decision · Program_Chairs · 2025-05-01

**Decision:**

Accept (poster)

**Comment:**

The manuscript received four high-quality reviews, the authors provided rebuttals, and the reviewers acknowledged and reflected on the rebuttals. None of the reviewers came to a clear negative or positive verdict, the average is slightly positive. The most negative reviewer M9Pw criticizes in particular the insufficient experiments, but acknowledges not being an expert in the field. Otherwise, the majority of comments point out a good but somehow limited contribution and a correct proof with potential detailed issues regarding the complexity calculation. Although the experimental results could be even more convincing, the manuscript is accepted.